# PICore: Physics-Informed Unsupervised Coreset Selection for Data Efficient Neural Operator Training

**Anirudh Satheesh**  *anirudhs@terpmail.umd.edu*
*Department of Computer Science*
*University of Maryland, College Park*

**Anant Khandelwal**  *akhandelwal79@gatech.edu*
*Georgia Institute of Technology*

**Mucong Ding**  *mcding@umd.edu*
*Department of Computer Science*
*University of Maryland, College Park*

**Radu Balan**  *rvbalan@umd.edu*
*Department of Mathematics*
*University of Maryland, College Park*

**Reviewed on OpenReview:** *https://openreview.net/forum?id=l0VSewTJCI*

## Abstract

Neural operators offer a powerful paradigm for solving partial differential equations (PDEs) that cannot be solved analytically by learning mappings between function spaces. However, there are two main bottlenecks in training neural operators: they require a significant amount of training data to learn these mappings, and this data needs to be labeled, which can only be accessed via expensive simulations with numerical solvers. To alleviate both of these issues simultaneously, we propose PICore, an unsupervised coreset selection framework that identifies the most informative training samples without requiring access to ground-truth PDE solutions. PICore leverages a physics-informed loss to select unlabeled inputs by their potential contribution to operator learning. After selecting a compact subset of inputs, only those samples are simulated using numerical solvers to generate labels, reducing annotation costs. We then train the neural operator on the reduced labeled dataset, significantly decreasing training time as well. Across four diverse PDE benchmarks and multiple coreset selection strategies, PICore achieves up to 78% average increase in training efficiency relative to supervised coreset selection methods with minimal changes in accuracy.

## 1 Introduction

Partial differential equations (PDEs) are foundational to modeling complex physical systems across science and engineering, from fluid dynamics to quantum mechanics. Most PDEs are non-analytic and need to be solved numerically via Finite Difference Methods (FDMs), Finite Element Methods (FEMs), and Finite Volume Methods (FVMs) Cyrus et al. (1968); Johnson (1988); Eriksson & Johnson (1995); LeVeque (2002). However, while these approaches yield high accuracy, they are computationally expensive because they require a simulation to be run to obtain a solution. This is especially true for high-resolution or multi-resolution PDEs, where simulations need to be re-run for each resolution.

Operator learning has emerged as a tool for accelerating PDE solutions by developing data-driven approximations using neural networks instead of traditional grid-based discretizations. Neural operators (Kovachki et al., 2023) are a family of neural networks that learn mappings between function spaces, such as initial conditions to solutions, which allows for resolution-invariant predictions. Models such as Fourier Neural

Operator (FNO) (Li et al., 2020) and U-Net Neural Operator (UNO) (Rahman et al., 2023) have shown state-of-the-art performance on various PDE benchmarks, and the ability to generalize to higher-order resolutions with minimal performance drops. Additional work, such as Physics Informed Neural Operator (PINO) (Li et al., 2024c) and Markov Neural Operator (MNO) (Li et al., 2021b), incorporates additional losses into neural operator training to improve performance and increase convergence speed.

Despite these advantages, there are two main data limitations of neural operators. First, they require significant amounts of training data to learn these mappings. Since PDE solvers require high-resolution data over several time frames for accurate training, such training data can be several gigabytes large (Takamoto et al., 2022). This poses a challenge for training in resource-constrained systems where such models would be trained and deployed, such as for weather prediction (Pathak et al., 2022; Bonev et al., 2023) and carbon storage (Tang et al., 2024). Secondly, this training data needs to be labeled by including both the initial condition and the ground truth solution. While generating initial conditions is cheap, as they can usually be sampled from a prior distribution, generating ground truth data requires running the full simulation through numerical solvers.

Coreset selection (Agarwal et al., 2005; Sener & Savarese, 2017) is a data-efficient training strategy that identifies a subset of the original training data that is most informative for model learning. Once this subset is identified, training only needs to be done on this subset, significantly reducing training time. However, this requires the full labeled training data to select a subset, which does not alleviate the cost of collecting labels. On the other hand, active learning (Gu et al., 2021; Cao & Tsang, 2022) minimizes data annotation costs by only labeling a subset of the training data at each iteration. Active learning selects a subset by a proxy metric such as Bayesian (Zhao et al., 2021; Beluch et al., 2018) or representation-based methods (Yang & Loog, 2022; Kim & Shin, 2022) at each training iteration, and trains only on that subset. A limitation of many iterative active learning strategies is that repeatedly alternating between selecting points and updating the model can increase training time and reduce convergence speed. Thus, we pose the following research question:

*How can we simultaneously reduce training time and labeling ground-truth solutions for Neural Operator learning?*

We address this problem using unsupervised coreset selection by identifying the most informative training samples based on the physics-informed loss (Li et al., 2024c), a criterion that does not require any ground truth labels. Our approach can also be viewed as a single-shot active learning implementation, where a subset of points is selected in one pass rather than iteratively. By leveraging this loss, we can prioritize samples likely to improve model performance without the need for expensive simulations. Ground truth labels are then generated only for this selected subset, significantly reducing the overall annotation cost. Finally, we train neural operator models on the reduced, high-quality dataset, leading to faster training times without compromising accuracy.

Our contributions are outlined as follows:

- **We propose PICore, a novel unsupervised framework that uniquely integrates physics-informed losses with coreset selection.** PICore eliminates the need for expensive ground-truth simulations during the data selection phase, simultaneously addressing the data annotation and training bottlenecks in neural operator training.

- **We demonstrate the modularity and generality of the PICore framework.** Our method is not tied to a specific architecture or selection algorithm, and we show its effectiveness across two different neural operators (FNO and UNO) and five distinct coreset selection strategies.

- **We present the first comprehensive benchmark for coreset selection in the context of neural operator learning.** Through extensive experiments on four diverse PDE datasets, we show that PICore achieves competitive accuracy to supervised methods while improving end-to-end training efficiency by up to 78% relative to supervised coreset selection.

## 2 Related Work

### 2.1 Neural Operators

While typical deep neural nets are used to map and model finite-dimensional vector spaces, such as text embeddings or images, neural operators map infinite-dimensional vector spaces, such as the space of functions (Kovachki et al., 2021). Neural operators are then widely used to represent differential equation solutions due to their ability to have a *family* of solutions. In the context of solving partial differential equations, a neural operator can take a function as an input (e.g. temperature at a point) and output a related function (e.g. heat over time at a point).

Among the first modern neural operators, DeepONet (Lu et al., 2021) uses the universal approximation theorem for operators with a branch and trunk network to model inputs and outputs. The Fourier Neural Operator (FNO) (Li et al., 2020) expands on this by performing kernel operations in Fourier space, which results in a more expressive model with better performance on more challenging PDE datasets, such as Navier Stokes. U-Net Neural Operator (UNO) (Rahman et al., 2023) expands on FNO by using a U-Net based structure to build deeper neural operators, and Convolutional Neural Operator (CNO) (Raonic et al., 2023) leverages convolutions to preserve the continuous structure of PDEs, even when discretized. Additional work improves training by incorporating additional losses. Physics Informed Neural Operator (PINO) (Li et al., 2024c) uses the physics informed loss to anchor the output to conform to the PDE dynamics, and Markov Neural Operator (MNO) (Li et al., 2022) uses dissipativity regularization to improve accuracy for more chaotic systems.

### 2.2 Data Efficient Machine Learning

#### 2.2.1 Coreset Selection

For problems where training is too expensive or slow, coreset selection can accelerate training while preserving accuracy. Coreset selection methods can be largely categorized into two types: training-free methods that leverage the geometric properties of the data, and training-based methods that use model-specific information to score data points. Training-free methods involve random (Guo et al., 2022; Gupta et al., 2023) and geometry-informed selection (Welling, 2009; Chen et al., 2012). Recent work on training-based methods can be split into three groups: (i) submodular approaches to maximize the coverage of the selected dataset (Wei et al., 2015; Mirzasoleiman et al., 2020; Pooladzandi et al., 2022), (ii) gradient-based approaches to exactly find the influence of a data point (Killamsetty et al., 2021a; Paul et al., 2021), and (iii) bilevel optimization methods to improve generalization performance (Killamsetty et al., 2021c;b).

The traditional testbed for coreset selection algorithms has been image classification tasks, but it also has applications in Neural Architecture Search (NAS) (Shim et al., 2021), efficient GAN training (Sinha et al., 2020), continual learning (Yoon et al., 2021), and large language model (LLM) finetuning (Zhang et al., 2025). However, to the best of our knowledge, coreset selection has not been used for improving the training efficiency of neural operator learning.

#### 2.2.2 Active Learning

In contrast to coreset selection, active learning, over multiple iterations and in an unsupervised environment, chooses previously unannotated data to label and trains on those newly labeled pairs (Li et al., 2025). The key differences are that active learning is unsupervised, choosing training samples with only features and that active learning is done over many iterations instead of in a single shot. Many algorithms transfer from coreset selection to active learning. Aside from the equivalent random selection, there are cluster based methods for active learning to find representative and typical examples (Sener & Savarese, 2017; Hacohen et al., 2022) and uncertainty based methods that find data for which the model is either uncertain or degraded (Rahmati et al., 2024; Houlsby et al., 2011; Ash et al., 2020). Wu et al. (2023) and Musekamp et al. (2025) show several uncertainty based sampling methods and active learning methods for physics informed learning, but these are limited to PINNs and not to neural operators, which are more powerful due to their ability to map between function spaces, but are inherently more difficult to perform active learning.

### 2.3 Active Learning and Data Efficiency for PDE Surrogates

Several works have explored improving data efficiency for PDE surrogate modeling through pretraining or architectural constraints. The closest to our approach is Chen et al. (2024), who introduce an unsupervised pretraining strategy using Masked Autoencoders (MAEs) to learn useful representations that are later fine-tuned with a smaller labeled dataset. While this reduces labeling requirements, it does so through a two-stage training pipeline. In contrast, PICore directly reduces both training cost and data annotation cost within a single training cycle. Hemmasian & Farimani (2024) reduce the cost of generating high-resolution training data by pretraining neural operators on lower-dimensional problems. This approach, however, relies on a factorized architecture such as the Factorized Fourier Neural Operator (FFNO) (Tran et al., 2021), whereas PICore is operator-architecture agnostic and can be applied to a wide range of neural operator models.

Another related line of work is active learning for PDE surrogate modeling. Li et al. (2024b) propose an active learning strategy tailored to FNOs, selecting new simulations by maximizing a utility–cost ratio. This method effectively reduces simulation labeling cost but does not address end-to-end training efficiency. Similarly, Kim et al. (2025) use a surrogate model to perform active learning at the timestep level, which improves efficiency only locally in time. In contrast, PICore reduces the annotation cost of full solution trajectories in a single pass. Beyond these architecture-specific approaches, Musekamp et al. (2025) study active learning for PINNs and Neural Operators more generally by evaluates several acquisition functions aimed at improving operator generalization under limited data. Overall, while prior work has made progress toward reducing labeling burden or improving sample efficiency, existing methods either rely on multi-stage pipelines, architectural constraints, or heuristic based active sampling. PICore differs by providing an architecture-agnostic, physics-informed coreset selection framework that improves both training efficiency and labeling cost in a unified manner.

## 3 Preliminaries

### 3.1 Neural Operators for PDE Solution Generation

Many physical systems can be modeled using partial differential equations (PDEs), which describe the evolution of a function $u \in \mathcal{U}$ over a domain. A general PDE can be expressed as

$$\mathcal{F}(u, a) = 0, \quad \text{on } \Omega \subset \mathbb{R}^d, \tag{1}$$

where $a \in \mathcal{A}$ represents input parameters such as boundary conditions, initial conditions, or physical coefficients; $\mathcal{F} : \mathcal{U} \times \mathcal{A} \to \mathcal{Z}$ is a differentiable and potentially nonlinear operator; and $\mathcal{A}, \mathcal{U}$ are Banach spaces over the bounded domain $\Omega$.

For stationary (time-independent) PDEs, the problem takes the form

$$\begin{aligned} \mathcal{F}(u, a) &= 0, \quad \text{on } \Omega \subset \mathbb{R}^d, \\ u &= h, \quad \text{on } \partial\Omega, \end{aligned} \tag{2}$$

where $h$ defines the boundary condition on the domain boundary $\partial\Omega$.

For dynamic (time-dependent) PDEs, the input $a$ is restricted to the initial condition $u|_{t=0}$, and the operator $\mathcal{F}$ is defined on the spatiotemporal domain $\Omega \times \mathcal{T}$:

$$\begin{aligned} \mathcal{F}(u, a) &= 0, \quad \text{on } \Omega \times \mathcal{T}, \\ u &= h, \quad \text{on } \partial\Omega \times \mathcal{T}, \\ u &= a, \quad \text{on } \Omega \times \{0\}, \end{aligned} \tag{3}$$

where $\mathcal{T} = (0, T)$ denotes the time domain. Examples of both stationary and dynamic PDEs are provided in Section A.

Unlike conventional neural networks that learn pointwise mappings, neural operators approximate solutions by learning mappings between infinite-dimensional function spaces:

$$\mathcal{G} : \mathcal{A} \to \mathcal{U}. \tag{4}$$

In practice, a PDE dataset consists of pairs $\{(a_i, u_i)\}_{i=1}^N$, where each $(a_i, u_i)$ corresponds to an input-output solution of the PDE. The neural operator $\mathcal{G}$ is approximated by $\mathcal{G}_\theta$ through the optimization

$$\mathcal{G}_\theta = \underset{\theta \in \Theta}{\arg\min} \frac{1}{N} \sum_{i=1}^N \|\mathcal{G}_\theta(a_i) - u_i\|_{L^2(\Omega)}^2, \tag{5}$$

where $\Theta$ is a finite-dimensional parameter space.

### 3.2 Coreset Selection

Given a dataset $D = \{(x_i, y_i)\}_{i=1}^N$, coreset selection aims to find a subset $S \subseteq D$ such that

$$S = \underset{S' \subset D, |S'| = \beta N}{\arg\min} \mathbb{E}_{(x_i, y_i) \sim S'}[\mathcal{L}(x_i, y_i; \theta^{S'})] \tag{6}$$

where $\beta$ is the percentage of the original dataset selected and $\theta^{S'}$ is the model trained on $S$. However, there are $\binom{N}{\beta N} = O(2^{NH_2(\beta)})$ possible subsets of size $\beta N$, so evaluating this objective directly is infeasible for large datasets. Instead, some works leverage a submodular function $f : 2^D \to \mathbb{R}$ which ensures the diminishing return property

$$f(S \cup \{z\}) - f(S) \geq f(T \cup \{z\}) - f(T), \quad \forall S \subseteq T \subseteq D, z \notin T \tag{7}$$

This results in a greedy selection procedure, significantly reducing the subset search space. Another way to perform coreset selection is to use a scoring function and select the top-$k$ data points. Finally, coreset selection can be represented as a bilevel optimization problem, resulting in the following form

$$S = \underset{S' \subset D, |S'| = \beta N}{\arg\min} \mathcal{L}(\theta^*(S')) \quad \text{s.t.} \quad \theta^*(S') = \underset{\theta \in \Theta}{\arg\min} \sum_{(x_i, y_i) \in S'} \mathcal{L}(x_i, y_i; \theta) \tag{8}$$

## 4 PICore

To address both issues of training time and data labeling costs for Neural Operator learning, we introduce PICore, an unsupervised coreset selection method that leverages a physics-informed loss to bypass the need for labeled training data during coreset selection.

Instead of using the ground truth PDE solution and supervised losses, the physics-informed loss evaluates the degree to which operator approximation $\mathcal{G}_\theta(a)$ satisfies the governing PDEs defined in either the stationary form or the dynamic form. The physics-informed loss penalizes violations of the PDE (PDE residual) in the interior of the domain, as well as deviations from the given boundary and initial conditions. For neural operators, the physics-informed loss is defined as

$$\mathcal{L}_{PI}(a; \theta) = \left\| \mathcal{F}(\mathcal{G}_\theta(a), a) \right\|_{L^2(\Omega)}^2 + \lambda \left\| \mathcal{G}_\theta(a)\big|_{\partial\Omega} - h \right\|_{L^2(\partial\Omega)}^2 \tag{9}$$

for stationary PDEs and

$$\mathcal{L}_{PI}(a; \theta) = \left\| \mathcal{F}(\mathcal{G}_\theta(a), a) \right\|_{L^2(\Omega \times \mathcal{T})}^2 + \lambda \left\| \mathcal{G}_\theta(a)\big|_{\partial\Omega \times \mathcal{T}} - h \right\|_{L^2(\partial\Omega \times \mathcal{T})}^2 + \mu \left\| \mathcal{G}_\theta(a)\big|_{t=0} - a \right\|_{L^2(\Omega)}^2 \tag{10}$$

for dynamic PDEs.

Given solely an unlabeled dataset $D = \{a_i\}_{i=1}^N$ that can be cheaply generated (usually by sampling from a prior distribution or sensor readings), PICore selects a coreset of $D$ by solving

$$S = \underset{S' \subset D, |S'| = \beta N}{\arg\min} \mathbb{E}_{a_i \sim S'}\left[ \mathcal{L}_{PI}\left( a_i; \theta^{S'} \right) \right] \tag{11}$$

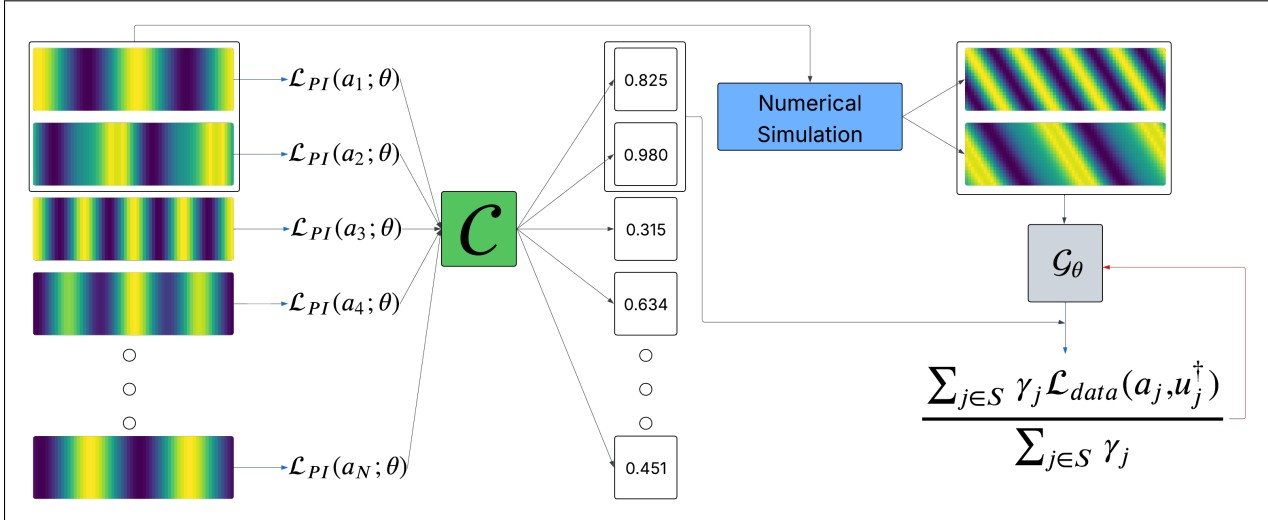

Figure 1: **Overview of the PICore Framework.** Given a set of initial conditions and a pre-trained (warm-started) neural operator $\mathcal{G}_\theta$, we compute the physics-informed loss $\mathcal{L}_{PI}(a_i; \theta)$ for each initial condition $a_i$. These losses are passed to a coreset selection algorithm $\mathcal{C}$, which identifies the most informative samples that deviate most from the underlying PDE. Each selected sample is assigned a weight $\gamma_j$ and is then simulated using a numerical solver to obtain the PDE solutions. The resulting labeled subset is used to update $\mathcal{G}_\theta$ using a weighted data loss, enabling efficient training by focusing on the most impactful data points. In the figure, blue arrows represent forward passes and red lines represent backward passes respectively.

using any existing coreset selection algorithm where $\theta^{S'}$ is the operator trained on $S'$. After selecting the coreset $S$, we simulate the true solutions $u_i^\dagger = \mathcal{G}(a_i)$ for each $a_i \in S$ using a traditional numerical solver, which forms the labeled subset $D_c = \{(a_i, u_i^\dagger)\}_{a_i \in S}$. Finally, we train the neural operator $\mathcal{G}_\theta$ on $D_c$ for $T$ epochs with the standard supervised data loss

$$\mathcal{L}_{\text{data}}(a_i, u_i^\dagger) = \|\mathcal{G}_\theta(a_i) - u_i^\dagger\|_{L^2(\Omega \times \mathcal{T})}^2 \tag{12}$$

Before coreset selection, we warm-start the neural operator with the physics-informed loss over the full dataset for a small number of epochs $T_w << T$. Warm starting is common in prior coreset selection methods (Killamsetty et al., 2021a) and is necessary as most coreset selection algorithms require gradient information, which is unusable with a randomly initialized model. We provide the full algorithm in Algorithm 1.

**Computing PDE Residuals** One challenge with using the physics-informed loss in coreset selection is computing the PDE residual $\mathcal{F}(\mathcal{G}_\theta(a), a)$. The residual requires computing derivatives of the neural operator with respect to the dimensional parameters, such as $\frac{\partial^2 \mathcal{G}_\theta}{\partial x \partial t}$. Li et al. (2024c) uses a function-wise differentiation method via Fourier differentiation to compute these values exactly, but this does not extend to a general class of neural operators. We also tried auto-differentiation methods, but these were highly computationally expensive, increasing the coreset selection time. Thus, we settled on simply using finite difference methods, which are efficient with linear time complexity in the input resolution.

## 5 Experimental Details

We conduct experiments on four representative PDE benchmarks spanning both stationary and time-dependent dynamics widely used in the neural operator literature:

- **1D Advection Equation** (time-dependent): A linear hyperbolic PDE representing pure transport dynamics, used to test propagation accuracy.

---

**Algorithm 1** PICore: Physics-Informed Coreset Selection for Neural Operators

---

**Require:** Unlabeled dataset $D = \{a_i\}_{i=1}^N$; coreset size $k = \beta N$; learning rate $\alpha$; pretrained operator $\mathcal{G}_\theta$; physics-informed loss $\mathcal{L}_{PI}(a; \theta)$; coreset selection algorithm $\mathcal{C}_{\text{select}}$; warmup steps $T_w$; training steps $T$

1: **Warm-start** $\mathcal{G}_\theta$ on unlabeled data using the physics-informed loss:
2: **for** $t = 1$ to $T_w$ **do**
3:     **for** each $a_i \in D$ **do**
4:         $\theta \leftarrow \theta - \alpha \nabla_\theta \mathcal{L}_{PI}(a_i; \theta)$
5:     **end for**
6: **end for**

7: **Score each sample** using physics-informed loss:
8: **for** each $a_i \in D$ **do**
9:     $\ell_i \leftarrow \mathcal{L}_{PI}(a_i; \theta)$
10: **end for**

11: **Select coreset indices** using $\mathcal{C}_{\text{select}}$:
12: $S \leftarrow \mathcal{C}_{\text{select}}(\{\ell_i\}_{i=1}^N, k)$

13: **Simulate ground truth for selected coreset:**
14: $D_c \leftarrow \emptyset$
15: **for** each $i \in S$ **do**
16:     $u_i^\dagger \leftarrow \mathcal{G}^\dagger(a_i)$ {Run numerical simulation}
17:     $D_c \leftarrow D_c \cup \{(a_i, u_i^\dagger)\}$
18: **end for**

19: **Train** $\mathcal{G}_\theta$ on $D_c$ using supervised loss:
20: **for** $t = 1$ to $T$ **do**
21:     **for** each $(a_i, u_i^\dagger) \in D_c$ **do**
22:         $\theta \leftarrow \theta - \alpha \nabla_\theta \mathcal{L}_{\text{data}}(a_i, u_i^\dagger; \theta)$
23:     **end for**
24: **end for**

---

- **1D Burgers' Equation** (time-dependent): A nonlinear convection-diffusion PDE with periodic boundary conditions, modeling shock formation and dissipation.

- **2D Darcy Flow** (stationary): A second-order elliptic PDE used to model pressure fields in porous media given heterogeneous permeability.

- **2D Navier-Stokes Incompressible Equation** (time-dependent): A nonlinear incompressible flow equation solved on a periodic domain.

Each dataset has 1000 generated trajectories, with 900 that can be used for training (varying based on the coreset selection percentage) and 100 for testing, which is comparable to existing neural operator literature (Li et al., 2021a; 2024c). We generate 20 timesteps forward for Advection and Burgers, and only 10 timesteps for the Navier Stokes Incompressible dataset due to memory limits. Additional information on the datasets can be found in Section A. We use the Fourier Neural Operator (FNO) (Li et al., 2020) and U-Net Neural Operator (Rahman et al., 2023) as the base models for all experiments due to their implementation simplicity and performance. However, PICore can work out of the box with any neural operator. We also use 5 coreset selection algorithms in our experiments: CRAIG (Mirzasoleiman et al., 2020), GradMatch (Killamsetty et al., 2021a), AdaCore (Pooladzandi et al., 2022), EL2N (Paul et al., 2021) and graNd (Paul et al., 2021). CRAIG, AdaCore, and GradMatch are submodular methods that try to match the gradient sum of the coreset to the gradient sum of the entire dataset. GraNd and EL2N are score based methods that use the gradient or the loss. Additional information on these coreset selection algorithms can be found in Section B. We use coreset selection percentages of 20%, 30%, 40%, 60%, and 80%. We report the results of each experiment with the

normalized root mean square error loss (NRMSE):

$$\frac{\|\mathcal{G}_\theta(a_i) - u_i^\dagger\|^2_{L^2(\Omega \times \mathcal{T})}}{\|u_i^\dagger\|^2_{L^2(\Omega \times \mathcal{T})}}$$

used in Takamoto et al. (2022). We use this as a normalized version of the data loss because the value of the $u_i^\dagger$ at each spatiotemporal point is very small, resulting in small MSE values and potential gradient vanishing during training. We also use the uniform spatiotemporal discretization at an input resolution of 64 for $\Omega$. Since FNO and UNO are resolution invariant, we also evaluate at higher resolutions for zero-shot super resolution in Section C.6. For all experiments we use $\lambda = 1$ and $\mu = 1$, but this is relatively arbitrary, we did not conduct any hyperparameter tuning.

We use $T_w = 25$ warmup epochs and reset the neural operator to its initialization to ensure fair comparisons between supervised and physics-informed coreset selection. Then, we train neural operators for $T = 500$ epochs and report the average NRMSE over 5 seeds on a held-out test set at the input resolution. We calculate the acceleration as the total time taken for supervised coreset selection / PICore (including data generation, warm starting, and training time) divided by the total time for the non-coreset baseline.

In addition to supervised coreset selection, we compare PICore to random subset selection, pure unsupervised training with the physics informed loss, and an active learning baseline based on uncertainty. Since most active learning baselines are for classification problems, we extend loss-as-uncertainty methods in Liu & Li (2023); Beluch et al. (2018) to neural operators. For the active learning baseline, we begin by randomly selecting 10% of the available data as an initial training set and generating the corresponding ground-truth PDE solutions. We then train 10 independent copies of the neural operator on this subset for $T_w$ epochs, which is the same training time for the other methods. After training, we construct the final coreset from the remaining unlabeled data in a single step by selecting the points exhibiting the highest variance across the model predictions.

## 6 Results

### 6.1 Main Results

We report the core findings for PICore and supervised coreset selection across the four representative PDE datasets in Tables 1, 2, 3, and 4. In addition to the average test NRMSE over the best coreset selection algorithm for each method, we show the decrease in full training time (including data annotation costs through simulation) relative to the non-coreset selection baseline. Our results demonstrate that PICore consistently achieves competitive test performance compared to supervised coreset selection while providing substantial computational efficiency gains, primarily by reducing expensive data annotation (simulation) costs during the coreset selection phase.

**PICore significantly improves training efficiency through reduced simulation costs.** Across four representative PDE datasets—Advection, Burgers, Darcy, and Navier-Stokes Incompressible—PICore consistently reduces the total training time by cutting down expensive simulation-based annotation. These efficiency gains become especially significant as the complexity of the PDE increases: Across the four datasets, PICore achieves average training time reductions of 0.9%, 9.8%, 30.1%, and 78.0% compared to supervised coreset selection, calculated by averaging the relative acceleration improvements at each selection percentage (20%, 30%, 40%, 60%, and 80%). For example, at a 20% coreset size, PICore achieves speedups of 5.01× on Darcy Flow (vs. 2.24× for supervised methods) and 5.00× on Navier-Stokes (vs. 1.14×) using UNO.

As shown in Tables 5, 6, 7, and 8, the relative contributions of training and data generation speedups vary by dataset difficulty. For simpler datasets such as Advection and Burgers, efficiency gains are driven primarily by reductions in training time. For example, Advection achieves a 79.7% improvement in training time but only a 1.47% improvement in data generation time at the 20% coreset level. In contrast, for more challenging datasets, the impact of training time reductions diminishes, while reductions in data generation time play a more significant role in overall efficiency gains. These results show that PICore scales well to high-dimensional scientific problems where data annotation costs dominate training.

**PICore matches supervised coreset methods in test accuracy at reduced data budgets.** Despite the substantial efficiency improvements, PICore remains competitive with strong supervised baselines in terms of test NRMSE. For instance, at a 20% coreset size, PICore achieves NRMSE values of $3.46 \times 10^{-2}$ for Advection using FNO and $2.84 \times 10^{-2}$ for Burgers using UNO, very close to the supervised method values of $3.42 \times 10^{-2}$ and $2.93 \times 10^{-2}$, respectively. This holds across different coreset sizes, with minor variations, showing that PICore can be as effective as supervised coreset selection, even at low coreset selection percentages. In fact, many dataset, model, and selection percentages combinations show that PICore improves upon supervised coreset selection, but there is largely no significant change in accuracy between the two methods. However, we observe that not all coreset selection algorithms perform equally as well. Due to the convexity assumptions and Hessian approximations with CRAIG (Mirzasoleiman et al., 2020) and AdaCore (Pooladzandi et al., 2022), they have higher NRMSE losses compared to the other algorithms. Thus, the tables almost always report GradMatch, GraNd, or El2N as the coreset algorithm to use for PICore and supervised coreset selection.

**Coreset Selection methods outperform Random and Active Learning baselines on most datasets.** Random subset selection consistently underperforms compared to both PICore and supervised coreset selection, with the performance gap widening as dataset complexity increases and the selection percentage decreases. For instance, on the Navier-Stokes Incompressible dataset at a 20% coreset size, random selection yields an nRMSE of $2.74 \times 10^{-1}$, whereas PICore achieves $1.12 \times 10^{-2}$ and supervised coreset selection achieves $9.57 \times 10^{-2}$. These results highlight that leveraging PDE-specific information, whether through supervised loss signals or physics-informed residuals, is crucial for attaining high-accuracy solutions with limited data. Nevertheless, the quality of the PDE residual approximation can introduce noise that diminishes performance, sometimes making PICore worse than random sampling on simpler problems such as Advection. In such cases, random sampling can reduce biases toward specific attributes of data points, thereby lowering errors attributable to those biases. We also observe that classical active learning occasionally outperforms PICore. For example, on the Advection dataset with UNO at medium coreset sizes (40–60%) and on the Navier-Stokes Incompressible dataset with FNO. However, across all other dataset-model combinations, active learning performs substantially worse, often by a wide margin.

**Physics-informed training alone is insufficient to achieve high accuracy.** Training a neural operator solely with the physics-informed loss on the full unlabeled dataset yields significantly worse performance than all other baselines, despite requiring no simulation cost. For example, on the Navier–Stokes Incompressible dataset with FNO, the physics-informed-only baseline achieves an nRMSE of $1.46 \times 10^{-0}$, compared to PICore's $1.25 \times 10^{-1}$ at 20% coreset selection, a decrease of an order of magnitude. This large gap arises because the physics-informed loss is unstable as a standalone training objective and fails to capture fine-grained solution details without supervised guidance. In contrast, PICore leverages the physics-informed loss as a proxy for selecting informative samples, then trains on a small labeled subset with the supervised nRMSE loss, which mitigates the physics-informed loss instability.

**There is a tradeoff between efficiency and absolute test accuracy.** While PICore offers strong performance and efficiency, one tradeoff is that the absolute test accuracy relative to training on 100% of the data is lower. For example, on the Advection dataset with FNO, the 100% training baseline yields an NRMSE of $2.13 \times 10^{-2}$, while PICore at 20% yields $3.77 \times 10^{-2}$. However, this is an inherent tradeoff for all coreset selection algorithms, as the selected coreset simply contains less information for training. Additionally, this is not specific to PICore, as similar reductions in accuracy hold for supervised coreset selection. In practice, one may want to select a higher selection percentage, such as 40%, which would yield higher accuracy ($2.69 \times 10^{-2}$) while still maintaining a competitive efficiency gain ($2.54\times$).

## 6.2 Further Results

In addition to investigating the efficacy of PICore, we also aim to answer the following questions: (1) How does PICore compare to existing unsupervised dataset selection methods? (2) How different are subsets selected by PICore compared to those selected by supervised coreset selection?

Table 1: Advection NRMSE at resolution 64

| Operator | Method | 20.0% | 30.0% | 40.0% | 60.0% | 80.0% | 100.0% |
|---|---|---|---|---|---|---|---|
| FNO | Physics-Informed | $8.43 \pm 0.03 \times 10^{-1}$ $(4.72\times)$ | $8.57 \pm 0.03 \times 10^{-1}$ $(3.06\times)$ | $8.73 \pm 0.04 \times 10^{-1}$ $(2.38\times)$ | $8.98 \pm 0.02 \times 10^{-1}$ $(1.59\times)$ | $9.14 \pm 0.02 \times 10^{-1}$ $(1.19\times)$ | $9.26 \pm 0.03 \times 10^{-1}$ $(0.96\times)$ |
| | Random | $\mathbf{3.39} \pm 0.07 \times 10^{-2}$ $(5.10\times)$ | $\mathbf{2.89} \pm 0.03 \times 10^{-2}$ $(3.32\times)$ | $\underline{2.68} \pm 0.02 \times 10^{-2}$ $(2.56\times)$ | $2.47 \pm 0.03 \times 10^{-2}$ $(1.72\times)$ | $2.37 \pm 0.04 \times 10^{-2}$ $(1.28\times)$ | $2.22 \pm 0.05 \times 10^{-2}$ $(1.00\times)$ |
| | Active Learning | $8.32 \pm 0.58 \times 10^{-2}$ $(5.04\times)$ | $6.29 \pm 0.40 \times 10^{-2}$ $(3.28\times)$ | $4.78 \pm 0.27 \times 10^{-2}$ $(2.52\times)$ | $3.51 \pm 0.16 \times 10^{-2}$ $(1.69\times)$ | $2.96 \pm 0.07 \times 10^{-2}$ $(1.26\times)$ | $2.22 \pm 0.05 \times 10^{-2}$ $(1.00\times)$ |
| | Supervised (graNd) | $\underline{3.42} \pm 0.12 \times 10^{-2}$ $(4.70\times)$ | $\underline{2.96} \pm 0.09 \times 10^{-2}$ $(3.15\times)$ | $\mathbf{2.64} \pm 0.03 \times 10^{-2}$ $(2.45\times)$ | $\underline{2.42} \pm 0.03 \times 10^{-2}$ $(1.66\times)$ | $\underline{2.25} \pm 0.02 \times 10^{-2}$ $(1.26\times)$ | $2.22 \pm 0.05 \times 10^{-2}$ $(1.00\times)$ |
| | PICore (graNd) | $3.46 \pm 0.13 \times 10^{-2}$ $(5.06\times)$ | $3.04 \pm 0.05 \times 10^{-2}$ $(3.27\times)$ | $2.69 \pm 0.05 \times 10^{-2}$ $(2.54\times)$ | $\mathbf{2.40} \pm 0.04 \times 10^{-2}$ $(1.68\times)$ | $\mathbf{2.25} \pm 0.04 \times 10^{-2}$ $(1.26\times)$ | $2.22 \pm 0.05 \times 10^{-2}$ $(1.00\times)$ |
| UNO | Physics-Informed | $8.07 \pm 0.05 \times 10^{-1}$ $(4.87\times)$ | $8.19 \pm 0.06 \times 10^{-1}$ $(3.21\times)$ | $8.28 \pm 0.05 \times 10^{-1}$ $(2.46\times)$ | $8.46 \pm 0.18 \times 10^{-1}$ $(1.65\times)$ | $9.09 \pm 0.37 \times 10^{-1}$ $(1.23\times)$ | $9.25 \pm 0.30 \times 10^{-1}$ $(0.98\times)$ |
| | Random | $1.59 \pm 0.02 \times 10^{-1}$ $(5.08\times)$ | $1.50 \pm 0.01 \times 10^{-1}$ $(3.35\times)$ | $1.44 \pm 0.007 \times 10^{-1}$ $(2.55\times)$ | $1.42 \pm 0.12 \times 10^{-1}$ $(1.70\times)$ | $1.32 \pm 0.12 \times 10^{-1}$ $(1.28\times)$ | $7.27 \pm 0.28 \times 10^{-2}$ $(1.00\times)$ |
| | Active Learning | $1.96 \pm 0.04 \times 10^{-1}$ $(5.05\times)$ | $1.59 \pm 0.05 \times 10^{-1}$ $(3.32\times)$ | $\mathbf{9.20} \pm 0.73 \times 10^{-2}$ $(2.52\times)$ | $\mathbf{7.49} \pm 0.47 \times 10^{-2}$ $(1.68\times)$ | $\mathbf{6.83} \pm 0.05 \times 10^{-2}$ $(1.26\times)$ | $7.27 \pm 0.28 \times 10^{-2}$ $(1.00\times)$ |
| | Supervised (gradmatch) | $\mathbf{1.55} \pm 0.02 \times 10^{-1}$ $(4.84\times)$ | $\underline{1.48} \pm 0.01 \times 10^{-1}$ $(3.23\times)$ | $\underline{1.42} \pm 0.02 \times 10^{-1}$ $(2.47\times)$ | $\underline{1.17} \pm 0.14 \times 10^{-1}$ $(1.67\times)$ | $\underline{8.69} \pm 1.23 \times 10^{-2}$ $(1.25\times)$ | $7.27 \pm 0.28 \times 10^{-2}$ $(1.00\times)$ |
| | PICore (gradmatch) | $\mathbf{1.55} \pm 0.01 \times 10^{-1}$ $(5.07\times)$ | $\mathbf{1.47} \pm 0.01 \times 10^{-1}$ $(3.34\times)$ | $1.43 \pm 0.008 \times 10^{-1}$ $(2.53\times)$ | $1.26 \pm 0.09 \times 10^{-1}$ $(1.69\times)$ | $9.06 \pm 1.07 \times 10^{-2}$ $(1.26\times)$ | $7.27 \pm 0.28 \times 10^{-2}$ $(1.00\times)$ |

Table 2: Burgers NRMSE at resolution 64

| Operator | Method | 20.0% | 30.0% | 40.0% | 60.0% | 80.0% | 100.0% |
|---|---|---|---|---|---|---|---|
| FNO | Physics-Informed | $5.26 \pm 0.12 \times 10^{-1}$ $(5.36\times)$ | $4.55 \pm 0.06 \times 10^{-1}$ $(3.45\times)$ | $4.36 \pm 0.07 \times 10^{-1}$ $(2.70\times)$ | $4.23 \pm 0.03 \times 10^{-1}$ $(1.80\times)$ | $4.19 \pm 0.03 \times 10^{-1}$ $(1.34\times)$ | $4.12 \pm 0.01 \times 10^{-1}$ $(1.07\times)$ |
| | Random | $1.85 \pm 0.09 \times 10^{-2}$ $(5.07\times)$ | $1.18 \pm 0.06 \times 10^{-2}$ $(3.31\times)$ | $8.23 \pm 0.21 \times 10^{-3}$ $(2.56\times)$ | $5.82 \pm 0.21 \times 10^{-3}$ $(1.72\times)$ | $4.75 \pm 0.13 \times 10^{-3}$ $(1.28\times)$ | $3.95 \pm 0.10 \times 10^{-3}$ $(1.00\times)$ |
| | Active Learning | $8.76 \pm 2.52 \times 10^{-2}$ $(5.03\times)$ | $4.57 \pm 0.95 \times 10^{-2}$ $(3.25\times)$ | $3.30 \pm 0.57 \times 10^{-2}$ $(2.52\times)$ | $2.06 \pm 0.07 \times 10^{-2}$ $(1.68\times)$ | $1.37 \pm 0.18 \times 10^{-2}$ $(1.26\times)$ | $3.95 \pm 0.10 \times 10^{-3}$ $(1.00\times)$ |
| | Supervised (gradmatch) | $\mathbf{1.71} \pm 0.16 \times 10^{-2}$ $(3.28\times)$ | $\underline{1.12} \pm 0.09 \times 10^{-2}$ $(2.52\times)$ | $\mathbf{7.68} \pm 0.29 \times 10^{-3}$ $(2.11\times)$ | $\mathbf{5.24} \pm 0.14 \times 10^{-3}$ $(1.55\times)$ | $\underline{4.13} \pm 0.08 \times 10^{-3}$ $(1.22\times)$ | $3.95 \pm 0.10 \times 10^{-3}$ $(1.00\times)$ |
| | PICore (el2n) | $\underline{1.81} \pm 0.08 \times 10^{-2}$ $(5.05\times)$ | $\mathbf{1.12} \pm 0.07 \times 10^{-2}$ $(3.30\times)$ | $\underline{8.07} \pm 0.33 \times 10^{-3}$ $(2.53\times)$ | $\underline{5.49} \pm 0.08 \times 10^{-3}$ $(1.68\times)$ | $\mathbf{4.07} \pm 0.10 \times 10^{-3}$ $(1.26\times)$ | $3.95 \pm 0.10 \times 10^{-3}$ $(1.00\times)$ |
| UNO | Physics-Informed | $4.82 \pm 0.07 \times 10^{-1}$ $(5.28\times)$ | $4.56 \pm 0.06 \times 10^{-1}$ $(3.46\times)$ | $4.48 \pm 0.06 \times 10^{-1}$ $(2.66\times)$ | $4.61 \pm 0.03 \times 10^{-1}$ $(1.77\times)$ | $4.63 \pm 0.02 \times 10^{-1}$ $(1.33\times)$ | $4.65 \pm 0.04 \times 10^{-1}$ $(1.06\times)$ |
| | Random | $\underline{2.92} \pm 0.05 \times 10^{-2}$ $(4.99\times)$ | $2.55 \pm 0.05 \times 10^{-2}$ $(3.34\times)$ | $2.25 \pm 0.05 \times 10^{-2}$ $(2.55\times)$ | $1.83 \pm 0.03 \times 10^{-2}$ $(1.70\times)$ | $1.58 \pm 0.01 \times 10^{-2}$ $(1.27\times)$ | $1.49 \pm 0.04 \times 10^{-2}$ $(1.00\times)$ |
| | Active Learning | $5.42 \pm 0.41 \times 10^{-2}$ $(5.01\times)$ | $4.12 \pm 0.11 \times 10^{-2}$ $(3.30\times)$ | $3.62 \pm 0.20 \times 10^{-2}$ $(2.51\times)$ | $3.03 \pm 0.23 \times 10^{-2}$ $(1.68\times)$ | $2.51 \pm 0.13 \times 10^{-2}$ $(1.26\times)$ | $1.49 \pm 0.04 \times 10^{-2}$ $(1.00\times)$ |
| | Supervised (gradmatch) | $2.93 \pm 0.10 \times 10^{-2}$ $(3.77\times)$ | $\underline{2.42} \pm 0.06 \times 10^{-2}$ $(2.77\times)$ | $\underline{2.08} \pm 0.05 \times 10^{-2}$ $(2.23\times)$ | $\underline{1.73} \pm 0.03 \times 10^{-2}$ $(1.59\times)$ | $\mathbf{1.54} \pm 0.02 \times 10^{-2}$ $(1.23\times)$ | $1.49 \pm 0.04 \times 10^{-2}$ $(1.00\times)$ |
| | PICore (graNd) | $\mathbf{2.84} \pm 0.05 \times 10^{-2}$ $(5.05\times)$ | $\mathbf{2.36} \pm 0.05 \times 10^{-2}$ $(3.33\times)$ | $\mathbf{2.06} \pm 0.04 \times 10^{-2}$ $(2.52\times)$ | $\mathbf{1.72} \pm 0.05 \times 10^{-2}$ $(1.69\times)$ | $\underline{1.57} \pm 0.03 \times 10^{-2}$ $(1.26\times)$ | $1.49 \pm 0.04 \times 10^{-2}$ $(1.00\times)$ |

### 6.2.1 Unsupervised Coreset Selection

We compare PICore to three unsupervised coreset selection methods: k-means clustering, cosine similarity, and Herding (Chen et al., 2012). For k-means clustering we use $k = \beta N$ clusters, and choose the data points closest to those clusters. For cosine similarity, we evaluate the cosine similarity between all pairs of points, and perform greedy selection to choose the coreset. We report direct comparison of the test NRMSE for both methods in Figures 2, 3, 4, and 5 in Section C.1. The results show that PICore consistently matches or outperforms the unsupervised baselines across all tested coreset sizes (20% to 80%) and neural operator architectures (FNO and UNO). For instance, on the Advection dataset at 20% coreset size, PICore with the EL2N algorithm achieves a test NRMSE of $3.29 \times 10^{-2}$, outperforming cosine similarity $3.39 \times 10^{-2}$ and herding $3.46 \times 10^{-2}$. Similar patterns are observed on the other datasets, indicating that PICore's selection strategy generalizes well across both time-dependent and stationary PDEs compared to other unsupervised coreset selection strategies. We also note that these trends hold across neural operator architectures, with PICore outperforming unsupervised methods with both FNO and UNO architectures. While FNO does consistently outperform UNO across datasets (except Navier Stokes Incompressible), this is due to the architecture differences and not due to PICore, as shown by the increase in NRMSE for UNO on the non-coreset baseline.

These results suggest that incorporating PDE-specific information can provide benefits over generic unsupervised selection methods. While clustering and similarity-based approaches provide reasonable coverage of

Table 3: Darcy NRMSE at resolution 64

| Operator | Method | 20.0% | 30.0% | 40.0% | 60.0% | 80.0% | 100.0% |
|---|---|---|---|---|---|---|---|
| FNO | Physics-Informed | $1.46 \pm 0.003 \times 10^0$ (7.43×) | $1.47 \pm 0.003 \times 10^0$ (4.96×) | $1.47 \pm 0.002 \times 10^0$ (3.75×) | $1.47 \pm 0.001 \times 10^0$ (2.50×) | $1.47 \pm 0.002 \times 10^0$ (1.88×) | $1.48 \pm 0.001 \times 10^0$ (1.51×) |
| | Random | $1.34 \pm 0.03 \times 10^{-1}$ (5.00×) | $1.15 \pm 0.01 \times 10^{-1}$ (3.36×) | $9.99 \pm 0.11 \times 10^{-2}$ (2.53×) | $7.94 \pm 0.04 \times 10^{-2}$ (1.69×) | $7.07 \pm 0.16 \times 10^{-2}$ (1.27×) | $6.18 \pm 0.09 \times 10^{-2}$ (1.00×) |
| | Active Learning | $2.01 \pm 0.16 \times 10^{-1}$ (4.99×) | $1.58 \pm 0.08 \times 10^{-1}$ (3.31×) | $1.25 \pm 0.06 \times 10^{-1}$ (2.50×) | $8.94 \pm 0.33 \times 10^{-2}$ (1.66×) | $7.19 \pm 0.25 \times 10^{-2}$ (1.25×) | $6.18 \pm 0.09 \times 10^{-2}$ (1.00×) |
| | Supervised (el2n) | $\underline{1.26} \pm 0.01 \times 10^{-1}$ (1.98×) | $\mathbf{1.07} \pm 0.007 \times 10^{-1}$ (1.76×) | $\mathbf{9.43} \pm 0.09 \times 10^{-2}$ (1.59×) | $\underline{7.83} \pm 0.18 \times 10^{-2}$ (1.33×) | $\mathbf{6.59} \pm 0.09 \times 10^{-2}$ (1.14×) | $6.18 \pm 0.09 \times 10^{-2}$ (1.00×) |
| | PICore (el2n) | $\mathbf{1.25} \pm 0.02 \times 10^{-1}$ (5.00×) | $\underline{1.12} \pm 0.02 \times 10^{-1}$ (3.32×) | $\underline{9.44} \pm 0.12 \times 10^{-2}$ (2.50×) | $\mathbf{7.77} \pm 0.18 \times 10^{-2}$ (1.67×) | $\underline{6.84} \pm 0.18 \times 10^{-2}$ (1.25×) | $6.18 \pm 0.09 \times 10^{-2}$ (1.00×) |
| UNO | Physics-Informed | $1.42 \pm 0.002 \times 10^0$ (7.05×) | $1.42 \pm 0.003 \times 10^0$ (4.71×) | $1.42 \pm 0.003 \times 10^0$ (3.55×) | $1.43 \pm 0.001 \times 10^0$ (2.37×) | $1.42 \pm 0.002 \times 10^0$ (1.78×) | $1.43 \pm 0.003 \times 10^0$ (1.46×) |
| | Random | $1.45 \pm 0.02 \times 10^{-1}$ (5.03×) | $1.22 \pm 0.03 \times 10^{-1}$ (3.37×) | $1.10 \pm 0.02 \times 10^{-1}$ (2.53×) | $9.23 \pm 0.22 \times 10^{-2}$ (1.69×) | $8.78 \pm 0.30 \times 10^{-2}$ (1.27×) | $7.57 \pm 0.13 \times 10^{-2}$ (1.00×) |
| | Active Learning | $1.87 \pm 0.14 \times 10^{-1}$ (5.04×) | $1.54 \pm 0.08 \times 10^{-1}$ (3.35×) | $1.27 \pm 0.06 \times 10^{-1}$ (2.52×) | $1.02 \pm 0.03 \times 10^{-1}$ (1.68×) | $8.63 \pm 0.19 \times 10^{-2}$ (1.26×) | $7.57 \pm 0.13 \times 10^{-2}$ (1.00×) |
| | Supervised (gradmatch) | $\mathbf{1.28} \pm 0.03 \times 10^{-1}$ (2.23×) | $\underline{1.14} \pm 0.01 \times 10^{-1}$ (1.93×) | $\underline{9.84} \pm 0.16 \times 10^{-2}$ (1.71×) | $\underline{8.60} \pm 0.11 \times 10^{-2}$ (1.38×) | $\underline{7.70} \pm 0.10 \times 10^{-2}$ (1.16×) | $7.57 \pm 0.13 \times 10^{-2}$ (1.00×) |
| | PICore (graNd) | $\mathbf{1.28} \pm 0.03 \times 10^{-1}$ (5.01×) | $\mathbf{1.12} \pm 0.01 \times 10^{-1}$ (3.33×) | $\mathbf{9.67} \pm 0.16 \times 10^{-2}$ (2.50×) | $\mathbf{8.42} \pm 0.14 \times 10^{-2}$ (1.67×) | $\mathbf{7.61} \pm 0.11 \times 10^{-2}$ (1.25×) | $7.57 \pm 0.13 \times 10^{-2}$ (1.00×) |

Table 4: Navier Stokes Incompressible NRMSE at resolution 64

| Operator | Method | 20.0% | 30.0% | 40.0% | 60.0% | 80.0% | 100.0% |
|---|---|---|---|---|---|---|---|
| FNO | Physics-Informed | $1.01 \pm 0.001 \times 10^0$ (69.41×) | $1.01 \pm 0.002 \times 10^0$ (46.30×) | $1.02 \pm 0.002 \times 10^0$ (34.73×) | $1.02 \pm 0.002 \times 10^0$ (23.29×) | $1.03 \pm 0.002 \times 10^0$ (17.43×) | $1.03 \pm 0.002 \times 10^0$ (13.32×) |
| | Random | $2.74 \pm 0.45 \times 10^{-1}$ (5.00×) | $5.59 \pm 0.80 \times 10^{-2}$ (3.34×) | $1.33 \pm 0.03 \times 10^{-2}$ (2.50×) | $9.06 \pm 0.24 \times 10^{-3}$ (1.67×) | $6.87 \pm 0.13 \times 10^{-3}$ (1.25×) | $5.66 \pm 0.11 \times 10^{-3}$ (1.00×) |
| | Active Learning | $\mathbf{9.32} \pm 6.96 \times 10^{-2}$ (5.00×) | $2.16 \pm 0.66 \times 10^{-2}$ (3.33×) | $1.27 \pm 0.03 \times 10^{-2}$ (2.50×) | $\mathbf{7.86} \pm 0.14 \times 10^{-3}$ (1.67×) | $\mathbf{6.24} \pm 0.21 \times 10^{-3}$ (1.25×) | $5.66 \pm 0.11 \times 10^{-3}$ (1.00×) |
| | Supervised (el2n) | $\underline{9.57} \pm 3.87 \times 10^{-2}$ (1.05×) | $\mathbf{1.75} \pm 0.07 \times 10^{-2}$ (1.05×) | $\mathbf{1.18} \pm 0.04 \times 10^{-2}$ (1.04×) | $\underline{7.94} \pm 0.18 \times 10^{-3}$ (1.03×) | $\underline{6.28} \pm 0.16 \times 10^{-3}$ (1.01×) | $5.66 \pm 0.11 \times 10^{-3}$ (1.00×) |
| | PICore (graNd) | $1.12 \pm 0.45 \times 10^{-1}$ (5.00×) | $\underline{1.81} \pm 0.12 \times 10^{-2}$ (3.33×) | $\underline{1.23} \pm 0.05 \times 10^{-2}$ (2.50×) | $8.00 \pm 0.23 \times 10^{-3}$ (1.67×) | $6.34 \pm 0.14 \times 10^{-3}$ (1.25×) | $5.66 \pm 0.11 \times 10^{-3}$ (1.00×) |
| UNO | Physics-Informed | $1.01 \pm 0.001 \times 10^0$ (29.42×) | $1.03 \pm 0.004 \times 10^0$ (19.61×) | $1.03 \pm 0.001 \times 10^0$ (14.74×) | $1.03 \pm 0.001 \times 10^0$ (9.86×) | $1.04 \pm 0.002 \times 10^0$ (7.38×) | $1.04 \pm 0.003 \times 10^0$ (5.89×) |
| | Random | $2.72 \pm 0.04 \times 10^{-2}$ (5.02×) | $2.24 \pm 0.02 \times 10^{-2}$ (3.34×) | $1.95 \pm 0.01 \times 10^{-2}$ (2.51×) | $1.61 \pm 0.006 \times 10^{-2}$ (1.67×) | $\mathbf{1.38} \pm 0.004 \times 10^{-2}$ (1.25×) | $1.24 \pm 0.004 \times 10^{-2}$ (1.00×) |
| | Active Learning | $2.91 \pm 0.05 \times 10^{-2}$ (5.00×) | $2.36 \pm 0.03 \times 10^{-2}$ (3.33×) | $2.06 \pm 0.02 \times 10^{-2}$ (2.50×) | $1.61 \pm 0.01 \times 10^{-2}$ (1.67×) | $1.40 \pm 0.009 \times 10^{-2}$ (1.25×) | $1.24 \pm 0.004 \times 10^{-2}$ (1.00×) |
| | Supervised (el2n) | $\underline{2.60} \pm 0.02 \times 10^{-2}$ (1.14×) | $\mathbf{2.19} \pm 0.02 \times 10^{-2}$ (1.12×) | $\underline{1.93} \pm 0.01 \times 10^{-2}$ (1.10×) | $\mathbf{1.57} \pm 0.009 \times 10^{-2}$ (1.07×) | $\mathbf{1.38} \pm 0.010 \times 10^{-2}$ (1.03×) | $1.24 \pm 0.004 \times 10^{-2}$ (1.00×) |
| | PICore (gradmatch) | $\mathbf{2.59} \pm 0.03 \times 10^{-2}$ (5.00×) | $\underline{2.20} \pm 0.009 \times 10^{-2}$ (3.33×) | $\mathbf{1.92} \pm 0.009 \times 10^{-2}$ (2.50×) | $\underline{1.60} \pm 0.005 \times 10^{-2}$ (1.67×) | $1.40 \pm 0.007 \times 10^{-2}$ (1.25×) | $1.24 \pm 0.004 \times 10^{-2}$ (1.00×) |

the input space, PICore's physics-informed selection offers a more targeted approach that identifies samples where the model violates physical constraints. This results in generally competitive or improved predictive accuracy, though the margin varies across datasets and selection percentages. For best results, practitioners should use El2N

### 6.2.2 Convergence of Coreset Selection vs Active Learning

A key distinction between coreset selection and active learning lies in their approach to data selection, which in turn affects their convergence speed. This iterative nature can be suboptimal, as it can lead to selecting redundant data points (Li et al., 2024a). While some works have shown superior convergence of active learning methods (Haimovich et al., 2024), these are under specific optimizer settings and in easier image classification domains. Our empirical results largely validate this viewpoint, demonstrating that PICore's single-shot selection generally leads to better subset selection that converges faster than active learning baselines. Figures 7 and 8 show the training loss convergence of PICore's coreset selection methods compared to the active learning baseline. For both FNO and UNO, the active learning method converges much slower by 2-3×. The difference in loss convergence decreases for more complex datasets such as Navier Stokes, but this is due to learning a larger FNO-3D / UNO-3D model than due to the subset selection method itself.

| Coreset % | FNO | | | UNO | | |
|---|---|---|---|---|---|---|
| | Train | Data | Warm-up | Train | Data | Warm-up |
| 20.0% | +79.7% | +1.4% | -0.9% | +80.4% | +0.8% | -4.5% |
| 30.0% | +69.7% | +1.2% | -1.4% | +70.7% | +0.7% | -4.5% |
| 40.0% | +61.3% | +1.1% | -1.8% | +61.7% | +0.6% | -4.5% |
| 60.0% | +42.8% | +0.7% | -2.8% | +43.2% | +0.4% | -4.5% |
| 80.0% | +24.2% | +0.4% | -3.7% | +24.2% | +0.2% | -4.5% |

Table 5: Advection PICore component speedup.

| Coreset % | FNO | | | UNO | | |
|---|---|---|---|---|---|---|
| | Train | Data | Warm-up | Train | Data | Warm-up |
| 20.0% | +70.3% | +10.7% | -0.8% | +74.4% | +6.6% | -4.5% |
| 30.0% | +61.5% | +9.4% | -1.3% | +65.4% | +5.8% | -4.5% |
| 40.0% | +54.0% | +8.1% | -1.6% | +57.1% | +5.0% | -4.5% |
| 60.0% | +37.7% | +5.4% | -2.4% | +39.9% | +3.3% | -4.5% |
| 80.0% | +21.4% | +2.7% | -3.3% | +22.3% | +1.7% | -4.5% |

Table 6: Burgers PICore component speedup.

| Coreset % | FNO | | | UNO | | |
|---|---|---|---|---|---|---|
| | Train | Data | Warm-up | Train | Data | Warm-up |
| 20.0% | +50.2% | +30.4% | -0.6% | +55.9% | +24.8% | -3.3% |
| 30.0% | +44.2% | +26.6% | -0.9% | +49.2% | +21.7% | -3.3% |
| 40.0% | +38.4% | +22.8% | -1.2% | +42.7% | +18.6% | -3.3% |
| 60.0% | +26.7% | +15.2% | -1.8% | +29.7% | +12.4% | -3.3% |
| 80.0% | +14.9% | +7.6% | -2.4% | +16.5% | +6.2% | -3.3% |

Table 7: Darcy PICore component speedup.

| Coreset % | FNO | | | UNO | | |
|---|---|---|---|---|---|---|
| | Train | Data | Warm-up | Train | Data | Warm-up |
| 20.0% | +5.2% | +74.9% | -0.1% | +12.7% | +67.5% | -0.8% |
| 30.0% | +4.5% | +65.5% | -0.1% | +11.2% | +59.0% | -0.8% |
| 40.0% | +3.9% | +56.2% | -0.1% | +9.7% | +50.6% | -0.8% |
| 60.0% | +2.7% | +37.4% | -0.2% | +6.7% | +33.7% | -0.8% |
| 80.0% | +1.5% | +18.7% | -0.2% | +3.7% | +16.9% | -0.8% |

Table 8: Navier Stokes PICore component speedup.

# 7 Limitations and Future Work

A key limitation of PICore is its reliance on a differentiable and known PDE to compute the physics-informed loss, which may not always be available. Practitioners can mitigate this by estimating the governing equations analytically using domain knowledge and trustworthy auxiliary models, or numerically through data-driven surrogates or weak-form formulations that approximate the PDE and its derivatives. Another limitation is PICore's dependence on existing coreset selection algorithms such as CRAIG and AdaCore, which were designed under convexity assumptions and use gradient or Hessian approximations. Their behavior on highly non-convex PDE learning landscapes is not well understood and can degrade performance, especially at low selection ratios. For instance, AdaCore with Hutchinson Hessian estimates applied only to the final layer consistently performs worse than alternative methods. In practice, we recommend GradMatch, EL2N, or graNd, which rely on fewer model-specific assumptions. Future work includes developing coreset selection methods tailored to neural operators by exploiting their inductive biases, and extending PICore to more general geometries to improve generalization while maintaining data efficiency.

# 8 Conclusion

In this work, we introduced PICore, a physics-informed unsupervised coreset selection framework designed to enhance the data efficiency of neural operator training. By leveraging the physics-informed loss to identify the most informative samples without requiring labeled data, PICore significantly reduces both the computational cost of numerical simulations and the time required for training. Our experiments across four PDE benchmarks demonstrate that PICore achieves competitive accuracy while reducing training costs by up to 78% compared to supervised coreset selection methods. Although PICore inherits some limitations from existing selection methods, we believe its ability to reduce labeling costs and accelerate training makes it a promising tool for large-scale scientific machine learning.

# 9 Acknowledgments

Satheesh and Balan have been supported in part by the National Science Foundation under grant DMS-2510216. Satheesh is also partially supported by a Daniel Sweet Undergraduate Research Fellowship ($2500). We thank Saptarashmi Bandyopadhyay and Stephen Casper for helpful feedback.

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

## A    PDE Datasets

For our experiments, we use several differential equation training sets to evaluate our algorithm. Each of these is used at an input grid resolution of 64. For the Advection, Burgers, and Darcy Flow equations, we generate datasets using code provided by Takamoto et al. (2022). For the Navier-Stokes Incompressible equation dataset, we generate data from Li et al. (2020).

### A.1    Advection

We construct our dataset by numerically solving the linear advection equation on the periodic domain $(0, 1)$:

$$\partial_t u(t, x) + \beta \, \partial_x u(t, x) = 0, \quad t \in (0, 2], \; x \in (0, 1), \tag{13}$$

The initial condition is defined as a superposition of sinusoidal modes,

$$u_0(x) \; = \; \sum_{i=1}^{N} A_i \, \sin\big(k_i x + \phi_i\big), \quad k_i = \frac{2\pi n_i}{L_x}, \tag{14}$$

where each $n_i$ is drawn uniformly from the range of integers from 1 to 8, $N$ is the number of waves, and the amplitudes $A_i \in [0, 1]$ and phases $\phi_i \in (0, 2\pi)$ are chosen at random. After assembly of $u_0(x)$, we apply with 10% probability each a pointwise absolute-value operation or multiplication by a smooth window function.

### A.2  Burger's Equation

We are interested in the one-dimensional viscous Burgers equation on the unit interval with periodic boundary conditions:

$$\partial_t u(t,x) + \partial_x\left(\tfrac{1}{2}\,u^2(t,x)\right) = \frac{\nu}{\pi}\,\partial_{xx} u(t,x), \quad x \in (0,1),\ t \in (0,2], \tag{15}$$

subject to the initial condition

$$u(0,x) = u_0(x), \quad x \in (0,1). \tag{16}$$

Here $\nu > 0$ is a constant diffusion coefficient. We use the nondimensional Reynolds number

$$R \;=\; \frac{\pi\,u_L}{\nu},$$

where $u_L$ is a characteristic velocity scale. In analogy with the Navier–Stokes equations, $R > 1$ indicates a regime dominated by nonlinear steepening and potential shock formation, whereas $R < 1$ corresponds to diffusion-dominated smooth dynamics.

### A.3  Darcy Flow

We obtain the steady-state solution of Darcy's equation on the unit square by evolving a time-dependent problem until convergence. The target elliptic problem is

$$-\nabla\cdot\left(a(x)\,\nabla u(x)\right) = f(x), \qquad\qquad x \in (0,1)^2, \tag{17}$$
$$u(x) = 0, \qquad\qquad x \in \partial(0,1)^2, \tag{18}$$

where $a(x)$ is the spatially varying coefficient and $f(x) \equiv \beta$ is a constant forcing that scales the solution amplitude.

Rather than solving equation 17, we integrate the parabolic problem

$$\partial_t u(x,t) - \nabla\cdot\left(a(x)\,\nabla u(x,t)\right) = \beta, \quad x \in (0,1)^2,\ t > 0, \tag{19}$$

with an appropriate random-field initial condition and homogeneous Dirichlet boundary data. We use the strong form $\nabla\cdot(a\nabla u) - f$ for the residual as in Li et al. (2024c).

### A.4  Navier-Stokes Equation

We consider the vorticity formulation on the periodic domain $(0,1)^2$:

$$\partial_t \omega + u\cdot\nabla\omega = \nu\,\Delta\omega + f, \quad \nabla\cdot u = 0, \quad \omega(x,0) \sim \mathcal{N}\left(0,\ 7^{3/2}(-\Delta + 49I)^{-2.5}\right),$$

with forcing

$$f(x) = 0.1\left[\sin 2\pi(x_1 + x_2) + \cos 2\pi(x_1 + x_2)\right].$$

The solution is obtained on a $256 \times 256$ grid via a Fourier pseudospectral scheme: first, we solve $\Delta\psi = -\omega$ in Fourier space to recover the stream function $\psi$ and velocity $u$, then compute the nonlinear advection term $u\cdot\nabla\omega$ in physical space with a 2/3-dealiasing filter, and finally advance in time using Crank–Nicolson for diffusion coupled with an explicit update for the nonlinear term.

## B  Coreset Selection Algorithms

In this section, we provide an overview of the coreset selection algorithms used. All implementations are our own, but are based on Guo et al. (2022).

**Adacore**

AdaCore augments CRAIG with second–order curvature so that difficult, high–influence samples are favoured even when first–order gradients look similar. In practice we **estimate only the diagonal** of the Hessian with *10 Hutchinson probes* per mini-batch, then pre-condition the last-layer gradient $\nabla \ell_i$ by element-wise division. Similarities are computed on these pre-conditioned vectors and the same stochastic-greedy routine as CRAIG is applied. The extra cost is the time to compute the approximation by deriving multiplications of the Hessian and arbitrary vectors via the Hessian-Free method (Yao et al., 2018), the time of Hutchinson's method to find the diagonal, and the time to apply the diagonal to the gradients of the last layer.

**EL2N**

Our EL2N (Error L2-Norm) coreset selection method follows from the premise that samples that are most worthwhile for the model have the highest losses. EL2N conducts a full training pass, where for each minibatch $x_i$, we calculate the loss without reduction for each individual sample, and calculate the norm for $x_i$'s loss vector. At the end of the epoch, we take the top $k$ minibatches by loss norm and return them with equal weight.

**CRAIG**

CRAIG (**C**oresets for **A**ccelerating **I**ncremental **G**radient-descent) selects a weighted subset of size $k$ whose gradients cover (i.e. represent) all per-example gradients. Let $g_i = \nabla_\theta \ell_i(\theta) \in \mathbb{R}^d$ be the gradient for example $i$. CRAIG finds a near optimal solution to the following problem.

$$A^* = \underset{A \subset V}{\arg \min} \, |S|, \; \sum_{n \in V} \min_{m \in S} \max_\theta ||g_n - g_m||$$

so every $g_i$ is "covered" by its most similar selected gradient. CRAIG selects the smallest subset S such that every example gradient is close (in $\mathcal{L}_2$) to at least one gradient in S. We approximate the coverage objective with the stochastic-greedy algorithm applied to the pairwise Euclidean similarity matrix of last-layer gradients. Greedy (or stochastic-greedy) selection gives a $(1 - 1/e)$-approximation in finite similarity evaluations. After $S$ is chosen, CRAIG sets integer weights

$$\gamma_j \; = \; \left| \{ \, i : \underset{m \in S}{\arg \max} \, s_{im} = j \} \right|, \quad j \in S,$$

so the weighted coreset gradient $\sum_{j \in S} \gamma_j g_j$ closely matches the full gradient $\sum_{i=1}^n g_i$ at each optimisation step. In practice the method is applied to last-layer gradients to reduce dimensionality without degrading the approximation quality.

**GradMatch**

Let the last–layer per-example gradients be concatenated as $A = [g_1 \, g_2 \, \ldots g_n] \in \mathbb{R}^{d \times n}$ and define the full-batch gradient $b = \frac{1}{n} \sum_{i=1}^n g_i$. GRADMATCH casts coreset selection as the sparse approximation problem.

$$\min_{x \in \mathbb{R}^n} \; \|Ax - b\|_2^2 \quad \text{s.t.} \quad \|x\|_0 \le k, \; x \ge 0.$$

OMP builds the weight vector $x$ greedily. Starting with residual $r = b$ and empty support $S$: (i) choose the column $j^\star = \arg \max_{j \notin S} A_j^\top r$; (ii) add $j^\star$ to $S$; (iii) refit the coefficients by non-negative least squares $x_S = \arg \min_{x \ge 0} \|A_S x - b\|_2^2 + \lambda \|x\|_2^2$; (iv) update $r = b - A_S x_S$. The loop terminates after $k$ selections, giving a coreset $S = \text{supp}(x)$ with weights $\gamma_j = x_j$.

During training we replace the full loss by the weighted loss $\sum_{j \in S} \gamma_j \ell_j / \sum_{j \in S} \gamma_j$, ensuring the mini-batch gradient of the coreset closely follows the full-batch gradient throughout optimisation.

**GraNd**

GraNd is similar to EL2N, but simply orders samples by the norm of their individual gradients and keeps the top $k$. We piggy-back on the same per-sample gradient collection already needed for CRAIG/GradMatch, but *stop after the first backward call.* We can rapidly sort these norms on the CPU, and use the selected indices for our coreset.

# C  Further Results

## C.1  Unsupervised Selection Methods Comparison

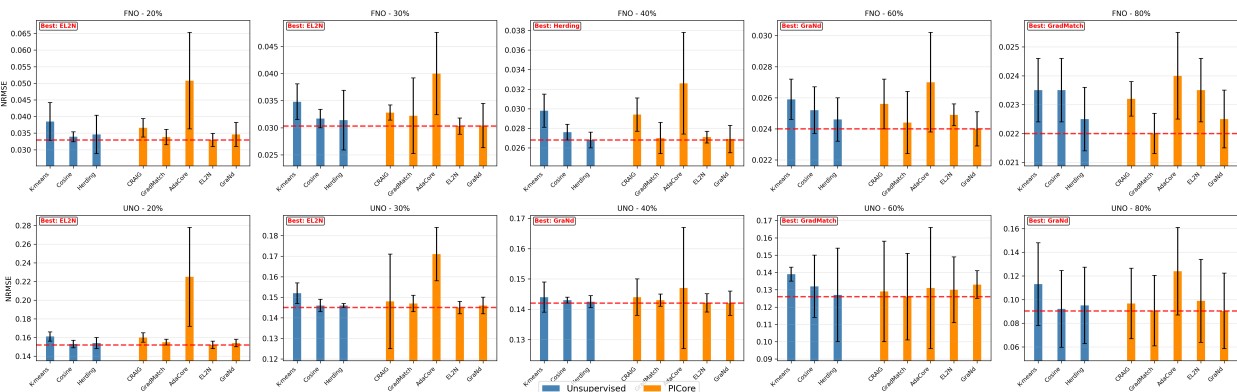

Figure 2: Test NRMSE on the Advection dataset at resolution 64 across varying coreset percentages (20%–100%) between unsupervised and PICore-based coreset selection methods using both FNO and UNO architectures.

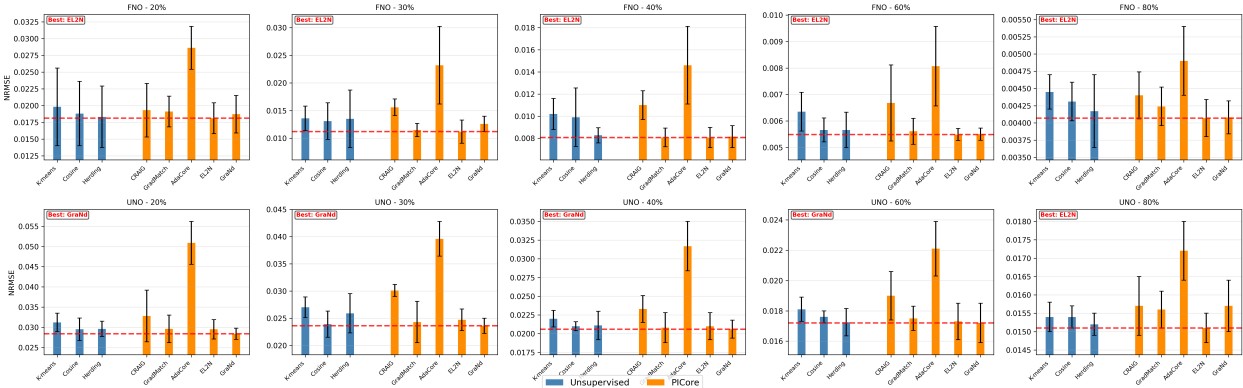

Figure 3: Test NRMSE on the Burgers dataset at resolution 64 across varying coreset percentages (20%–100%) between unsupervised and PICore-based coreset selection methods using both FNO and UNO architectures.

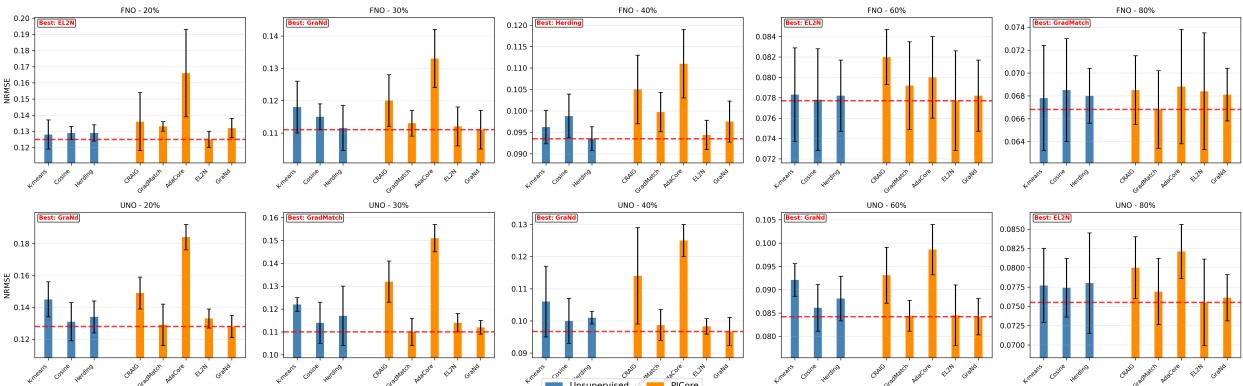

Figure 4: Test NRMSE on the Darcy dataset at resolution 64 across varying coreset percentages (20%–100%) between unsupervised and PICore-based coreset selection methods using both FNO and UNO architectures.

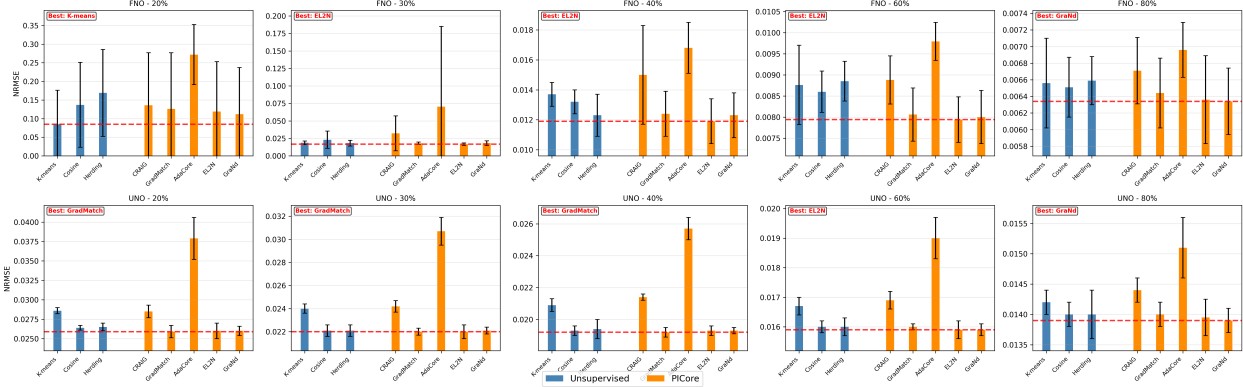

Figure 5: Test NRMSE on the Navier Stokes Incompressible dataset at resolution 64 across varying coreset percentages (20%–100%) between unsupervised and PICore-based coreset selection methods using both FNO and UNO architectures.

## C.2 Spatial Orientation of Supervised Coreset Selection and PICore

To better understand the differences between supervised coreset selection and PICore, we analyze how well each method covers the input space by computing the average distance from coreset points to their centroid, which serves as a proxy for spread or diversity. We compute this distance with respect to the $\|\cdot\|_{L^2(\Omega)}$ norm, where the centroid is the average data point element-wise and the average distance is the average norm between the centroid and the selected data points in the coreset. As shown in Figure 6, this distance is nearly identical across datasets and neural operators (FNO and UNO), with overlapping standard error bars with differences decreasing as the PDE complexity increases (Advection to Navier Stokes). This suggests that PICore selects coresets that are as well-distributed as those from supervised methods, despite not using labeled data. The comparable coverage indicates that differences in downstream performance likely arise from the type of points selected rather than their spatial distribution.

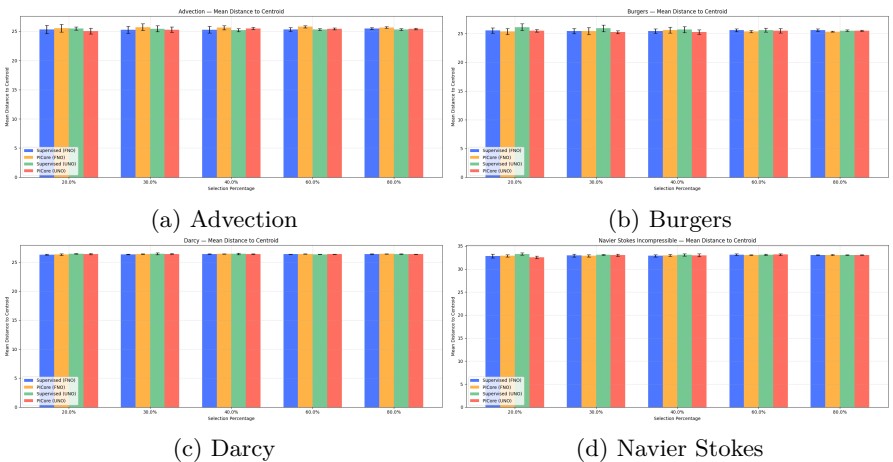

Figure 6: Average centroid distances across datasets for FNO and UNO.

## C.3 PICore Training Convergence

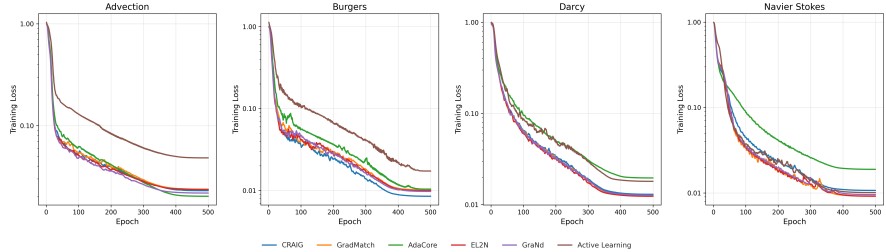

Figure 7: Training convergence of coreset selection v.s. active learning using FNO at a 20% selection ratio.

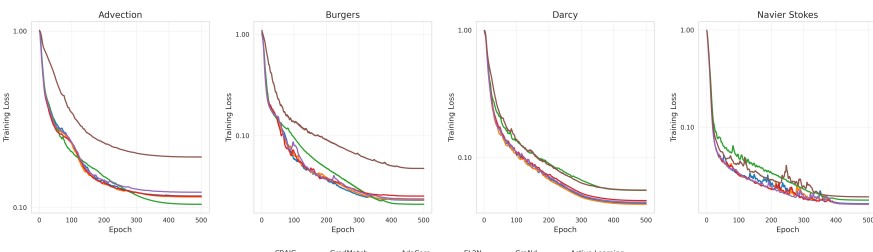

Figure 8: Training convergence of coreset selection v.s. active learning using UNO at a 20% selection ratio.

### C.4 Ablation Study

Table 9: Ablation on Warm Start at resolution 64

| Operator | $T_w$ | 20.0% | 30.0% | 40.0% | 60.0% | 80.0% | 100.0% |
|---|---|---|---|---|---|---|---|
| FNO | 10 epochs | $1.64 \pm 0.14 \times 10^{-2}$ $(3.23\times)$ | $1.14 \pm 0.10 \times 10^{-2}$ $(2.48\times)$ | $8.07 \pm 0.48 \times 10^{-3}$ $(2.06\times)$ | $5.32 \pm 0.30 \times 10^{-3}$ $(1.52\times)$ | $4.17 \pm 0.17 \times 10^{-3}$ $(1.19\times)$ | $3.95 \pm 0.10 \times 10^{-3}$ $(1.00\times)$ |
| | 25 epochs | $1.52 \pm 0.13 \times 10^{-2}$ $(3.18\times)$ | $1.07 \pm 0.06 \times 10^{-2}$ $(2.44\times)$ | $7.71 \pm 0.33 \times 10^{-3}$ $(2.02\times)$ | $5.33 \pm 0.14 \times 10^{-3}$ $(1.48\times)$ | $4.15 \pm 0.06 \times 10^{-3}$ $(1.17\times)$ | $3.95 \pm 0.10 \times 10^{-3}$ $(1.00\times)$ |
| | 50 epochs | $1.69 \pm 0.07 \times 10^{-2}$ $(3.10\times)$ | $1.12 \pm 0.08 \times 10^{-2}$ $(2.36\times)$ | $7.96 \pm 0.38 \times 10^{-3}$ $(1.95\times)$ | $5.51 \pm 0.17 \times 10^{-3}$ $(1.43\times)$ | $4.16 \pm 0.11 \times 10^{-3}$ $(1.13\times)$ | $3.95 \pm 0.10 \times 10^{-3}$ $(1.00\times)$ |
| | 100 epochs | $1.69 \pm 0.14 \times 10^{-2}$ $(2.94\times)$ | $1.06 \pm 0.04 \times 10^{-2}$ $(2.21\times)$ | $8.24 \pm 0.45 \times 10^{-3}$ $(1.83\times)$ | $5.40 \pm 0.16 \times 10^{-3}$ $(1.34\times)$ | $4.15 \pm 0.05 \times 10^{-3}$ $(1.05\times)$ | $3.95 \pm 0.10 \times 10^{-3}$ $(1.00\times)$ |
| UNO | 10 epochs | $2.97 \pm 0.04 \times 10^{-2}$ $(3.79\times)$ | $2.44 \pm 0.06 \times 10^{-2}$ $(2.79\times)$ | $2.13 \pm 0.04 \times 10^{-2}$ $(2.25\times)$ | $1.77 \pm 0.04 \times 10^{-2}$ $(1.60\times)$ | $1.57 \pm 0.01 \times 10^{-2}$ $(1.24\times)$ | $1.49 \pm 0.04 \times 10^{-2}$ $(1.00\times)$ |
| | 25 epochs | $2.99 \pm 0.04 \times 10^{-2}$ $(3.72\times)$ | $2.51 \pm 0.04 \times 10^{-2}$ $(2.73\times)$ | $2.09 \pm 0.02 \times 10^{-2}$ $(2.19\times)$ | $1.76 \pm 0.03 \times 10^{-2}$ $(1.56\times)$ | $1.57 \pm 0.04 \times 10^{-2}$ $(1.20\times)$ | $1.49 \pm 0.04 \times 10^{-2}$ $(1.00\times)$ |
| | 50 epochs | $2.89 \pm 0.03 \times 10^{-2}$ $(3.60\times)$ | $2.42 \pm 0.03 \times 10^{-2}$ $(2.64\times)$ | $2.18 \pm 0.05 \times 10^{-2}$ $(2.11\times)$ | $1.73 \pm 0.03 \times 10^{-2}$ $(1.49\times)$ | $1.54 \pm 0.02 \times 10^{-2}$ $(1.15\times)$ | $1.49 \pm 0.04 \times 10^{-2}$ $(1.00\times)$ |
| | 100 epochs | $3.00 \pm 0.08 \times 10^{-2}$ $(3.38\times)$ | $2.53 \pm 0.05 \times 10^{-2}$ $(2.46\times)$ | $2.05 \pm 0.04 \times 10^{-2}$ $(1.96\times)$ | $1.73 \pm 0.03 \times 10^{-2}$ $(1.38\times)$ | $1.55 \pm 0.02 \times 10^{-2}$ $(1.07\times)$ | $1.49 \pm 0.04 \times 10^{-2}$ $(1.00\times)$ |

The ablation study in Table 9 examines how different warm start epochs influence the performance of PICore on FNO and UNO. We fix the coreset selection algorithm to EL2N and the dataset to Burgers. Across warm start configurations, extending the number of epochs beyond 10 to 25 or 50 epochs leads to only marginal gains, and by 100 epochs, the improvements are negligible or even slightly inconsistent within the bounds of standard deviation. For FNO, the 25-epoch variant achieves the lowest errors at lower data fractions (20–40%), suggesting that moderate warm starting may yield slightly better initialization for PICore. UNO, on the other hand, shows stable performance across all pretraining lengths, implying that it benefits less from extended warm starting. Overall, the table suggests that minimal warm starting (around 10–25 epochs) is sufficient for both operator families, while additional finetuning offers little benefit.

### C.5 Multi-Resolution Coreset Selection

Table 10: Burgers NRMSE with Multi Resolution Data

| Operator | Method | Algorithm | 20.0% | 30.0% | 40.0% | 60.0% | 80.0% | 100.0% |
|---|---|---|---|---|---|---|---|---|
| FNO | Supervised | craig | $8.29 \pm 1.58 \times 10^{-2}$ | $7.73 \pm 0.83 \times 10^{-2}$ | $7.56 \pm 0.75 \times 10^{-2}$ | $7.78 \pm 0.88 \times 10^{-2}$ | $7.25 \pm 0.84 \times 10^{-2}$ | $6.85 \pm 0.79 \times 10^{-2}$ |
| | | gradmatch | $8.33 \pm 1.16 \times 10^{-2}$ | $8.10 \pm 0.93 \times 10^{-2}$ | $7.92 \pm 0.44 \times 10^{-2}$ | $7.64 \pm 0.81 \times 10^{-2}$ | $6.94 \pm 0.58 \times 10^{-2}$ | $6.85 \pm 0.79 \times 10^{-2}$ |
| | | adacore | $9.24 \pm 0.83 \times 10^{-2}$ | $8.73 \pm 0.54 \times 10^{-2}$ | $8.03 \pm 0.98 \times 10^{-2}$ | $7.44 \pm 0.93 \times 10^{-2}$ | $6.89 \pm 0.75 \times 10^{-2}$ | $6.85 \pm 0.79 \times 10^{-2}$ |
| | | el2n | $7.61 \pm 0.96 \times 10^{-2}$ | $\mathbf{7.34 \pm 1.30 \times 10^{-2}}$ | $7.55 \pm 0.60 \times 10^{-2}$ | $7.30 \pm 0.61 \times 10^{-2}$ | $6.94 \pm 0.85 \times 10^{-2}$ | $6.85 \pm 0.79 \times 10^{-2}$ |
| | | graNd | $8.44 \pm 0.73 \times 10^{-2}$ | $7.95 \pm 1.00 \times 10^{-2}$ | $7.75 \pm 0.88 \times 10^{-2}$ | $7.45 \pm 0.60 \times 10^{-2}$ | $7.23 \pm 0.58 \times 10^{-2}$ | $6.85 \pm 0.79 \times 10^{-2}$ |
| | | *Acceleration* | $3.75 \pm 0.00\times$ | $2.76 \pm 0.00\times$ | $2.24 \pm 0.00\times$ | $1.60 \pm 0.00\times$ | $1.24 \pm 0.00\times$ | $1.00 \pm 0.00\times$ |
| | PICore | craig | $\mathbf{7.37 \pm 1.63 \times 10^{-2}}$ | $8.09 \pm 0.62 \times 10^{-2}$ | $7.70 \pm 1.20 \times 10^{-2}$ | $7.36 \pm 0.70 \times 10^{-2}$ | $7.31 \pm 0.45 \times 10^{-2}$ | $6.85 \pm 0.79 \times 10^{-2}$ |
| | | gradmatch | $7.90 \pm 1.89 \times 10^{-2}$ | $8.07 \pm 1.55 \times 10^{-2}$ | $7.79 \pm 1.19 \times 10^{-2}$ | $7.54 \pm 0.59 \times 10^{-2}$ | $7.15 \pm 0.85 \times 10^{-2}$ | $6.85 \pm 0.79 \times 10^{-2}$ |
| | | adacore | $8.56 \pm 1.78 \times 10^{-2}$ | $8.14 \pm 1.36 \times 10^{-2}$ | $8.20 \pm 0.81 \times 10^{-2}$ | $7.90 \pm 1.03 \times 10^{-2}$ | $7.34 \pm 1.10 \times 10^{-2}$ | $6.85 \pm 0.79 \times 10^{-2}$ |
| | | el2n | $8.10 \pm 0.66 \times 10^{-2}$ | $7.37 \pm 1.87 \times 10^{-2}$ | $\mathbf{7.49 \pm 1.54 \times 10^{-2}}$ | $\mathbf{7.29 \pm 0.65 \times 10^{-2}}$ | $\mathbf{6.78 \pm 0.73 \times 10^{-2}}$ | $6.85 \pm 0.79 \times 10^{-2}$ |
| | | graNd | $8.88 \pm 0.63 \times 10^{-2}$ | $8.39 \pm 0.87 \times 10^{-2}$ | $8.19 \pm 1.16 \times 10^{-2}$ | $7.56 \pm 0.61 \times 10^{-2}$ | $7.31 \pm 0.81 \times 10^{-2}$ | $6.85 \pm 0.79 \times 10^{-2}$ |
| | | *Acceleration* | $5.08 \pm 0.01\times$ | $3.32 \pm 0.00\times$ | $2.54 \pm 0.00\times$ | $1.70 \pm 0.00\times$ | $1.27 \pm 0.00\times$ | $1.00 \pm 0.00\times$ |

Table 10 compares Supervised and Physics-Informed Coreset Selection (PICore) across varying data resolutions for the Burgers' equation using the FNO operator. In this experiment, the dataset is split into two equal parts: one half is generated at a resolution of 32, and the other at a resolution of 64, introducing multi-scale variability in the training data. Overall, PICore demonstrates improved stability and often superior accuracy at lower data percentages, particularly in the FNO setting. For instance, under FNO with 20–60% data, PICore achieves consistently lower NRMSE values than supervised methods, especially with the el2n and craig algorithms, indicating better generalization when fewer samples are available. This advantage reflects the integration of physical constraints, which act as a strong inductive bias that helps retain key solution structures even when data is sparse or noisy. The mixed-resolution setup further amplifies this effect, as

while supervised coresets can struggle to reconcile the differences between coarse (32) and fine (64) grids, PICore leverages physics-informed consistency to align features across scales, resulting in more stable and resolution-invariant representations of the Burgers' dynamics. However, this setup also introduces greater variability and noise, as discrepancies between resolutions can create inconsistencies in gradient magnitudes and feature smoothness, but these issues are generally inherent to multi-resolution datasets, rather than specific to the coreset selection approach itself.

## C.6 Zero Shot Super Resolution

Since Neural Operators learn parameters independently of the discretization (unlike PINNs), trained neural operators can perform zero-shot super-resolution, which allows for training a model at a lower resolution and evaluating at a higher resolution. We scale the Advection and Burgers datasets to resolutions of 128 and 256, the Darcy dataset to 128, and the Navier Stokes Incompressible dataset to size 256 in tables 11–16.

| Operator | Method | Algorithm | 20.0% | 30.0% | 40.0% | 60.0% | 80.0% | 100.0% |
|---|---|---|---|---|---|---|---|---|
| FNO | Supervised | craig | $7.69 \times 10^{-2}$ | $8.41 \times 10^{-2}$ | $8.05 \times 10^{-2}$ | $8.04 \times 10^{-2}$ | $7.61 \times 10^{-2}$ | $5.90 \times 10^{-2}$ |
| | | gradmatch | $8.38 \times 10^{-2}$ | $7.87 \times 10^{-2}$ | $7.85 \times 10^{-2}$ | $7.86 \times 10^{-2}$ | $7.73 \times 10^{-2}$ | $5.90 \times 10^{-2}$ |
| | | adacore | $8.45 \times 10^{-2}$ | $7.85 \times 10^{-2}$ | $7.99 \times 10^{-2}$ | $7.42 \times 10^{-2}$ | $7.43 \times 10^{-2}$ | $5.90 \times 10^{-2}$ |
| | | el2n | $8.34 \times 10^{-2}$ | $8.05 \times 10^{-2}$ | $8.10 \times 10^{-2}$ | $7.88 \times 10^{-2}$ | $7.67 \times 10^{-2}$ | $5.90 \times 10^{-2}$ |
| | | graNd | $8.25 \times 10^{-2}$ | $8.05 \times 10^{-2}$ | $7.79 \times 10^{-2}$ | $7.60 \times 10^{-2}$ | $7.26 \times 10^{-2}$ | $5.90 \times 10^{-2}$ |
| | PICore | craig | $8.38 \times 10^{-2}$ | $8.01 \times 10^{-2}$ | $7.76 \times 10^{-2}$ | $7.41 \times 10^{-2}$ | $7.35 \times 10^{-2}$ | $5.90 \times 10^{-2}$ |
| | | gradmatch | $8.85 \times 10^{-2}$ | $8.35 \times 10^{-2}$ | $8.15 \times 10^{-2}$ | $7.76 \times 10^{-2}$ | $7.59 \times 10^{-2}$ | $5.90 \times 10^{-2}$ |
| | | adacore | $8.82 \times 10^{-2}$ | $7.57 \times 10^{-2}$ | $7.28 \times 10^{-2}$ | $7.27 \times 10^{-2}$ | $7.03 \times 10^{-2}$ | $5.90 \times 10^{-2}$ |
| | | el2n | $8.95 \times 10^{-2}$ | $8.42 \times 10^{-2}$ | $8.33 \times 10^{-2}$ | $8.18 \times 10^{-2}$ | $7.98 \times 10^{-2}$ | $5.90 \times 10^{-2}$ |
| | | graNd | $8.38 \times 10^{-2}$ | $8.15 \times 10^{-2}$ | $7.86 \times 10^{-2}$ | $7.72 \times 10^{-2}$ | $7.50 \times 10^{-2}$ | $5.90 \times 10^{-2}$ |
| UNO | Supervised | craig | $1.72 \times 10^{-1}$ | $1.66 \times 10^{-1}$ | $1.61 \times 10^{-1}$ | $1.48 \times 10^{-1}$ | $1.14 \times 10^{-1}$ | $1.09 \times 10^{-1}$ |
| | | gradmatch | $1.70 \times 10^{-1}$ | $1.65 \times 10^{-1}$ | $1.61 \times 10^{-1}$ | $1.42 \times 10^{-1}$ | $1.21 \times 10^{-1}$ | $1.09 \times 10^{-1}$ |
| | | adacore | $2.25 \times 10^{-1}$ | $1.85 \times 10^{-1}$ | $1.71 \times 10^{-1}$ | $1.58 \times 10^{-1}$ | $1.27 \times 10^{-1}$ | $1.09 \times 10^{-1}$ |
| | | el2n | $1.69 \times 10^{-1}$ | $1.65 \times 10^{-1}$ | $1.62 \times 10^{-1}$ | $1.55 \times 10^{-1}$ | $1.25 \times 10^{-1}$ | $1.09 \times 10^{-1}$ |
| | | graNd | $1.69 \times 10^{-1}$ | $1.64 \times 10^{-1}$ | $1.61 \times 10^{-1}$ | $1.49 \times 10^{-1}$ | $1.24 \times 10^{-1}$ | $1.09 \times 10^{-1}$ |
| | PICore | craig | $1.74 \times 10^{-1}$ | $1.61 \times 10^{-1}$ | $1.60 \times 10^{-1}$ | $1.51 \times 10^{-1}$ | $1.26 \times 10^{-1}$ | $1.09 \times 10^{-1}$ |
| | | gradmatch | $1.70 \times 10^{-1}$ | $1.65 \times 10^{-1}$ | $1.62 \times 10^{-1}$ | $1.49 \times 10^{-1}$ | $1.23 \times 10^{-1}$ | $1.09 \times 10^{-1}$ |
| | | adacore | $2.32 \times 10^{-1}$ | $1.82 \times 10^{-1}$ | $1.61 \times 10^{-1}$ | $1.50 \times 10^{-1}$ | $1.45 \times 10^{-1}$ | $1.09 \times 10^{-1}$ |
| | | el2n | $1.69 \times 10^{-1}$ | $1.64 \times 10^{-1}$ | $1.63 \times 10^{-1}$ | $1.52 \times 10^{-1}$ | $1.27 \times 10^{-1}$ | $1.09 \times 10^{-1}$ |
| | | graNd | $1.70 \times 10^{-1}$ | $1.64 \times 10^{-1}$ | $1.62 \times 10^{-1}$ | $1.53 \times 10^{-1}$ | $1.23 \times 10^{-1}$ | $1.09 \times 10^{-1}$ |

Table 11: Test NRMSE on the Advection dataset at resolution 128 across varying coreset percentages (20%–100%) between supervised and PICore-based coreset selection methods using both FNO and UNO architectures.

| Operator | Method | Algorithm | 20.0% | 30.0% | 40.0% | 60.0% | 80.0% | 100.0% |
|---|---|---|---|---|---|---|---|---|
| FNO | Supervised | craig | $8.55 \times 10^{-2}$ | $9.30 \times 10^{-2}$ | $8.99 \times 10^{-2}$ | $9.00 \times 10^{-2}$ | $8.64 \times 10^{-2}$ | $7.07 \times 10^{-2}$ |
| | | gradmatch | $9.22 \times 10^{-2}$ | $8.79 \times 10^{-2}$ | $8.80 \times 10^{-2}$ | $8.85 \times 10^{-2}$ | $8.76 \times 10^{-2}$ | $7.07 \times 10^{-2}$ |
| | | adacore | $9.07 \times 10^{-2}$ | $8.65 \times 10^{-2}$ | $8.87 \times 10^{-2}$ | $8.43 \times 10^{-2}$ | $8.47 \times 10^{-2}$ | $7.07 \times 10^{-2}$ |
| | | el2n | $9.18 \times 10^{-2}$ | $8.98 \times 10^{-2}$ | $9.04 \times 10^{-2}$ | $8.85 \times 10^{-2}$ | $8.67 \times 10^{-2}$ | $7.07 \times 10^{-2}$ |
| | | graNd | $9.09 \times 10^{-2}$ | $8.95 \times 10^{-2}$ | $8.76 \times 10^{-2}$ | $8.59 \times 10^{-2}$ | $8.31 \times 10^{-2}$ | $7.07 \times 10^{-2}$ |
| | PICore | craig | $9.15 \times 10^{-2}$ | $8.86 \times 10^{-2}$ | $8.67 \times 10^{-2}$ | $8.39 \times 10^{-2}$ | $8.38 \times 10^{-2}$ | $7.07 \times 10^{-2}$ |
| | | gradmatch | $9.68 \times 10^{-2}$ | $9.28 \times 10^{-2}$ | $9.12 \times 10^{-2}$ | $8.79 \times 10^{-2}$ | $8.63 \times 10^{-2}$ | $7.07 \times 10^{-2}$ |
| | | adacore | $9.40 \times 10^{-2}$ | $8.37 \times 10^{-2}$ | $8.20 \times 10^{-2}$ | $8.27 \times 10^{-2}$ | $8.08 \times 10^{-2}$ | $7.07 \times 10^{-2}$ |
| | | el2n | $9.77 \times 10^{-2}$ | $9.30 \times 10^{-2}$ | $9.24 \times 10^{-2}$ | $9.13 \times 10^{-2}$ | $8.96 \times 10^{-2}$ | $7.07 \times 10^{-2}$ |
| | | graNd | $9.23 \times 10^{-2}$ | $9.06 \times 10^{-2}$ | $8.84 \times 10^{-2}$ | $8.77 \times 10^{-2}$ | $8.56 \times 10^{-2}$ | $7.07 \times 10^{-2}$ |
| UNO | Supervised | craig | $1.86 \times 10^{-1}$ | $1.80 \times 10^{-1}$ | $1.76 \times 10^{-1}$ | $1.63 \times 10^{-1}$ | $1.35 \times 10^{-1}$ | $1.31 \times 10^{-1}$ |
| | | gradmatch | $1.84 \times 10^{-1}$ | $1.79 \times 10^{-1}$ | $1.77 \times 10^{-1}$ | $1.59 \times 10^{-1}$ | $1.41 \times 10^{-1}$ | $1.31 \times 10^{-1}$ |
| | | adacore | $2.30 \times 10^{-1}$ | $1.95 \times 10^{-1}$ | $1.83 \times 10^{-1}$ | $1.73 \times 10^{-1}$ | $1.46 \times 10^{-1}$ | $1.31 \times 10^{-1}$ |
| | | el2n | $1.83 \times 10^{-1}$ | $1.80 \times 10^{-1}$ | $1.78 \times 10^{-1}$ | $1.71 \times 10^{-1}$ | $1.44 \times 10^{-1}$ | $1.31 \times 10^{-1}$ |
| | | graNd | $1.82 \times 10^{-1}$ | $1.79 \times 10^{-1}$ | $1.76 \times 10^{-1}$ | $1.64 \times 10^{-1}$ | $1.43 \times 10^{-1}$ | $1.31 \times 10^{-1}$ |
| | PICore | craig | $1.87 \times 10^{-1}$ | $1.74 \times 10^{-1}$ | $1.75 \times 10^{-1}$ | $1.67 \times 10^{-1}$ | $1.45 \times 10^{-1}$ | $1.31 \times 10^{-1}$ |
| | | gradmatch | $1.83 \times 10^{-1}$ | $1.80 \times 10^{-1}$ | $1.78 \times 10^{-1}$ | $1.65 \times 10^{-1}$ | $1.42 \times 10^{-1}$ | $1.31 \times 10^{-1}$ |
| | | adacore | $2.38 \times 10^{-1}$ | $1.92 \times 10^{-1}$ | $1.75 \times 10^{-1}$ | $1.66 \times 10^{-1}$ | $1.61 \times 10^{-1}$ | $1.31 \times 10^{-1}$ |
| | | el2n | $1.84 \times 10^{-1}$ | $1.79 \times 10^{-1}$ | $1.78 \times 10^{-1}$ | $1.68 \times 10^{-1}$ | $1.47 \times 10^{-1}$ | $1.31 \times 10^{-1}$ |
| | | graNd | $1.83 \times 10^{-1}$ | $1.79 \times 10^{-1}$ | $1.77 \times 10^{-1}$ | $1.68 \times 10^{-1}$ | $1.43 \times 10^{-1}$ | $1.31 \times 10^{-1}$ |

Table 12: Test NRMSE on the Advection dataset at resolution 256 across varying coreset percentages (20%–100%) between supervised and PICore-based coreset selection methods using both FNO and UNO architectures.

| Operator | Method | Algorithm | 20.0% | 30.0% | 40.0% | 60.0% | 80.0% | 100.0% |
|---|---|---|---|---|---|---|---|---|
| FNO | Supervised | craig | $6.83 \times 10^{-2}$ | $7.32 \times 10^{-2}$ | $7.20 \times 10^{-2}$ | $6.90 \times 10^{-2}$ | $6.14 \times 10^{-2}$ | $4.57 \times 10^{-2}$ |
| | | gradmatch | $6.56 \times 10^{-2}$ | $6.35 \times 10^{-2}$ | $6.68 \times 10^{-2}$ | $6.32 \times 10^{-2}$ | $6.36 \times 10^{-2}$ | $4.57 \times 10^{-2}$ |
| | | adacore | $6.15 \times 10^{-2}$ | $6.07 \times 10^{-2}$ | $6.72 \times 10^{-2}$ | $6.34 \times 10^{-2}$ | $6.28 \times 10^{-2}$ | $4.57 \times 10^{-2}$ |
| | | el2n | $6.86 \times 10^{-2}$ | $7.17 \times 10^{-2}$ | $6.74 \times 10^{-2}$ | $6.43 \times 10^{-2}$ | $6.08 \times 10^{-2}$ | $4.57 \times 10^{-2}$ |
| | | graNd | $6.72 \times 10^{-2}$ | $6.00 \times 10^{-2}$ | $6.47 \times 10^{-2}$ | $6.27 \times 10^{-2}$ | $6.05 \times 10^{-2}$ | $4.57 \times 10^{-2}$ |
| | PICore | craig | $6.67 \times 10^{-2}$ | $6.31 \times 10^{-2}$ | $6.38 \times 10^{-2}$ | $6.22 \times 10^{-2}$ | $6.39 \times 10^{-2}$ | $4.57 \times 10^{-2}$ |
| | | gradmatch | $6.40 \times 10^{-2}$ | $6.62 \times 10^{-2}$ | $6.81 \times 10^{-2}$ | $6.47 \times 10^{-2}$ | $6.15 \times 10^{-2}$ | $4.57 \times 10^{-2}$ |
| | | adacore | $7.39 \times 10^{-2}$ | $6.60 \times 10^{-2}$ | $6.59 \times 10^{-2}$ | $6.38 \times 10^{-2}$ | $6.46 \times 10^{-2}$ | $4.57 \times 10^{-2}$ |
| | | el2n | $6.29 \times 10^{-2}$ | $6.36 \times 10^{-2}$ | $6.60 \times 10^{-2}$ | $6.29 \times 10^{-2}$ | $6.38 \times 10^{-2}$ | $4.57 \times 10^{-2}$ |
| | | graNd | $6.71 \times 10^{-2}$ | $6.58 \times 10^{-2}$ | $6.91 \times 10^{-2}$ | $6.50 \times 10^{-2}$ | $6.45 \times 10^{-2}$ | $4.57 \times 10^{-2}$ |
| UNO | Supervised | craig | $2.94 \times 10^{-2}$ | $2.77 \times 10^{-2}$ | $2.62 \times 10^{-2}$ | $2.33 \times 10^{-2}$ | $2.19 \times 10^{-2}$ | $2.16 \times 10^{-2}$ |
| | | gradmatch | $3.00 \times 10^{-2}$ | $2.83 \times 10^{-2}$ | $2.67 \times 10^{-2}$ | $2.45 \times 10^{-2}$ | $2.31 \times 10^{-2}$ | $2.16 \times 10^{-2}$ |
| | | adacore | $4.36 \times 10^{-2}$ | $3.39 \times 10^{-2}$ | $2.98 \times 10^{-2}$ | $2.47 \times 10^{-2}$ | $2.23 \times 10^{-2}$ | $2.16 \times 10^{-2}$ |
| | | el2n | $2.94 \times 10^{-2}$ | $2.76 \times 10^{-2}$ | $2.62 \times 10^{-2}$ | $2.39 \times 10^{-2}$ | $2.28 \times 10^{-2}$ | $2.16 \times 10^{-2}$ |
| | | graNd | $3.01 \times 10^{-2}$ | $2.83 \times 10^{-2}$ | $2.69 \times 10^{-2}$ | $2.47 \times 10^{-2}$ | $2.30 \times 10^{-2}$ | $2.16 \times 10^{-2}$ |
| | PICore | craig | $3.18 \times 10^{-2}$ | $2.96 \times 10^{-2}$ | $2.77 \times 10^{-2}$ | $2.38 \times 10^{-2}$ | $2.15 \times 10^{-2}$ | $2.16 \times 10^{-2}$ |
| | | gradmatch | $2.97 \times 10^{-2}$ | $2.76 \times 10^{-2}$ | $2.69 \times 10^{-2}$ | $2.40 \times 10^{-2}$ | $2.27 \times 10^{-2}$ | $2.16 \times 10^{-2}$ |
| | | adacore | $4.77 \times 10^{-2}$ | $3.68 \times 10^{-2}$ | $3.09 \times 10^{-2}$ | $2.49 \times 10^{-2}$ | $2.19 \times 10^{-2}$ | $2.16 \times 10^{-2}$ |
| | | el2n | $2.96 \times 10^{-2}$ | $2.77 \times 10^{-2}$ | $2.62 \times 10^{-2}$ | $2.43 \times 10^{-2}$ | $2.29 \times 10^{-2}$ | $2.16 \times 10^{-2}$ |
| | | graNd | $2.89 \times 10^{-2}$ | $2.72 \times 10^{-2}$ | $2.66 \times 10^{-2}$ | $2.49 \times 10^{-2}$ | $2.29 \times 10^{-2}$ | $2.16 \times 10^{-2}$ |

Table 13: Test NRMSE on the Burgers dataset at resolution 128 across varying coreset percentages (20%–100%) between supervised and PICore-based coreset selection methods using both FNO and UNO architectures.

| Operator | Method | Algorithm | 20.0% | 30.0% | 40.0% | 60.0% | 80.0% | 100.0% |
|---|---|---|---|---|---|---|---|---|
| FNO | Supervised | craig | $7.02 \times 10^{-2}$ | $7.52 \times 10^{-2}$ | $7.40 \times 10^{-2}$ | $7.12 \times 10^{-2}$ | $6.37 \times 10^{-2}$ | $4.89 \times 10^{-2}$ |
| | | gradmatch | $6.72 \times 10^{-2}$ | $6.55 \times 10^{-2}$ | $6.89 \times 10^{-2}$ | $6.55 \times 10^{-2}$ | $6.60 \times 10^{-2}$ | $4.89 \times 10^{-2}$ |
| | | adacore | $6.26 \times 10^{-2}$ | $6.18 \times 10^{-2}$ | $6.90 \times 10^{-2}$ | $6.55 \times 10^{-2}$ | $6.52 \times 10^{-2}$ | $4.89 \times 10^{-2}$ |
| | | el2n | $7.04 \times 10^{-2}$ | $7.38 \times 10^{-2}$ | $6.96 \times 10^{-2}$ | $6.66 \times 10^{-2}$ | $6.32 \times 10^{-2}$ | $4.89 \times 10^{-2}$ |
| | | graNd | $6.89 \times 10^{-2}$ | $6.21 \times 10^{-2}$ | $6.69 \times 10^{-2}$ | $6.50 \times 10^{-2}$ | $6.29 \times 10^{-2}$ | $4.89 \times 10^{-2}$ |
| | PICore | craig | $6.85 \times 10^{-2}$ | $6.52 \times 10^{-2}$ | $6.59 \times 10^{-2}$ | $6.45 \times 10^{-2}$ | $6.63 \times 10^{-2}$ | $4.89 \times 10^{-2}$ |
| | | gradmatch | $6.57 \times 10^{-2}$ | $6.83 \times 10^{-2}$ | $7.02 \times 10^{-2}$ | $6.70 \times 10^{-2}$ | $6.39 \times 10^{-2}$ | $4.89 \times 10^{-2}$ |
| | | adacore | $7.50 \times 10^{-2}$ | $6.75 \times 10^{-2}$ | $6.80 \times 10^{-2}$ | $6.61 \times 10^{-2}$ | $6.69 \times 10^{-2}$ | $4.89 \times 10^{-2}$ |
| | | el2n | $6.49 \times 10^{-2}$ | $6.57 \times 10^{-2}$ | $6.82 \times 10^{-2}$ | $6.52 \times 10^{-2}$ | $6.62 \times 10^{-2}$ | $4.89 \times 10^{-2}$ |
| | | graNd | $6.88 \times 10^{-2}$ | $6.78 \times 10^{-2}$ | $7.12 \times 10^{-2}$ | $6.72 \times 10^{-2}$ | $6.69 \times 10^{-2}$ | $4.89 \times 10^{-2}$ |
| UNO | Supervised | craig | $3.38 \times 10^{-2}$ | $3.36 \times 10^{-2}$ | $3.26 \times 10^{-2}$ | $3.06 \times 10^{-2}$ | $2.96 \times 10^{-2}$ | $2.92 \times 10^{-2}$ |
| | | gradmatch | $3.52 \times 10^{-2}$ | $3.43 \times 10^{-2}$ | $3.36 \times 10^{-2}$ | $3.22 \times 10^{-2}$ | $3.10 \times 10^{-2}$ | $2.92 \times 10^{-2}$ |
| | | adacore | $4.70 \times 10^{-2}$ | $3.84 \times 10^{-2}$ | $3.51 \times 10^{-2}$ | $3.17 \times 10^{-2}$ | $3.01 \times 10^{-2}$ | $2.92 \times 10^{-2}$ |
| | | el2n | $3.46 \times 10^{-2}$ | $3.33 \times 10^{-2}$ | $3.27 \times 10^{-2}$ | $3.11 \times 10^{-2}$ | $3.05 \times 10^{-2}$ | $2.92 \times 10^{-2}$ |
| | | graNd | $3.51 \times 10^{-2}$ | $3.40 \times 10^{-2}$ | $3.34 \times 10^{-2}$ | $3.22 \times 10^{-2}$ | $3.11 \times 10^{-2}$ | $2.92 \times 10^{-2}$ |
| | PICore | craig | $3.63 \times 10^{-2}$ | $3.47 \times 10^{-2}$ | $3.36 \times 10^{-2}$ | $3.11 \times 10^{-2}$ | $2.97 \times 10^{-2}$ | $2.92 \times 10^{-2}$ |
| | | gradmatch | $3.50 \times 10^{-2}$ | $3.35 \times 10^{-2}$ | $3.34 \times 10^{-2}$ | $3.14 \times 10^{-2}$ | $3.08 \times 10^{-2}$ | $2.92 \times 10^{-2}$ |
| | | adacore | $5.08 \times 10^{-2}$ | $4.06 \times 10^{-2}$ | $3.57 \times 10^{-2}$ | $3.12 \times 10^{-2}$ | $2.95 \times 10^{-2}$ | $2.92 \times 10^{-2}$ |
| | | el2n | $3.47 \times 10^{-2}$ | $3.36 \times 10^{-2}$ | $3.29 \times 10^{-2}$ | $3.16 \times 10^{-2}$ | $3.09 \times 10^{-2}$ | $2.92 \times 10^{-2}$ |
| | | graNd | $3.41 \times 10^{-2}$ | $3.34 \times 10^{-2}$ | $3.34 \times 10^{-2}$ | $3.28 \times 10^{-2}$ | $3.12 \times 10^{-2}$ | $2.92 \times 10^{-2}$ |

Table 14: Test NRMSE on the Burgers dataset at resolution 256 across varying coreset percentages (20%–100%) between supervised and PICore-based coreset selection methods using both FNO and UNO architectures.

| Operator | Method | Algorithm | 20.0% | 30.0% | 40.0% | 60.0% | 80.0% | 100.0% |
|---|---|---|---|---|---|---|---|---|
| FNO | Supervised | craig | $1.67 \times 10^{-1}$ | $1.51 \times 10^{-1}$ | $1.36 \times 10^{-1}$ | $1.15 \times 10^{-1}$ | $1.02 \times 10^{-1}$ | $8.76 \times 10^{-2}$ |
| | | gradmatch | $1.72 \times 10^{-1}$ | $1.47 \times 10^{-1}$ | $1.36 \times 10^{-1}$ | $1.15 \times 10^{-1}$ | $1.04 \times 10^{-1}$ | $8.76 \times 10^{-2}$ |
| | | adacore | $1.88 \times 10^{-1}$ | $1.61 \times 10^{-1}$ | $1.41 \times 10^{-1}$ | $1.18 \times 10^{-1}$ | $1.05 \times 10^{-1}$ | $8.76 \times 10^{-2}$ |
| | | el2n | $1.68 \times 10^{-1}$ | $1.42 \times 10^{-1}$ | $1.33 \times 10^{-1}$ | $1.15 \times 10^{-1}$ | $1.02 \times 10^{-1}$ | $8.76 \times 10^{-2}$ |
| | | graNd | $1.67 \times 10^{-1}$ | $1.46 \times 10^{-1}$ | $1.32 \times 10^{-1}$ | $1.16 \times 10^{-1}$ | $1.04 \times 10^{-1}$ | $8.76 \times 10^{-2}$ |
| | PICore | craig | $1.68 \times 10^{-1}$ | $1.53 \times 10^{-1}$ | $1.39 \times 10^{-1}$ | $1.16 \times 10^{-1}$ | $1.03 \times 10^{-1}$ | $8.76 \times 10^{-2}$ |
| | | gradmatch | $1.73 \times 10^{-1}$ | $1.48 \times 10^{-1}$ | $1.38 \times 10^{-1}$ | $1.16 \times 10^{-1}$ | $1.03 \times 10^{-1}$ | $8.76 \times 10^{-2}$ |
| | | adacore | $1.98 \times 10^{-1}$ | $1.58 \times 10^{-1}$ | $1.37 \times 10^{-1}$ | $1.10 \times 10^{-1}$ | $1.02 \times 10^{-1}$ | $8.76 \times 10^{-2}$ |
| | | el2n | $1.66 \times 10^{-1}$ | $1.48 \times 10^{-1}$ | $1.32 \times 10^{-1}$ | $1.13 \times 10^{-1}$ | $1.03 \times 10^{-1}$ | $8.76 \times 10^{-2}$ |
| | | graNd | $1.72 \times 10^{-1}$ | $1.47 \times 10^{-1}$ | $1.35 \times 10^{-1}$ | $1.13 \times 10^{-1}$ | $1.03 \times 10^{-1}$ | $8.76 \times 10^{-2}$ |
| UNO | Supervised | craig | $1.36 \times 10^{0}$ | $1.57 \times 10^{0}$ | $1.57 \times 10^{0}$ | $1.48 \times 10^{0}$ | $1.40 \times 10^{0}$ | $1.72 \times 10^{0}$ |
| | | gradmatch | $1.65 \times 10^{0}$ | $1.57 \times 10^{0}$ | $1.69 \times 10^{0}$ | $1.74 \times 10^{0}$ | $1.55 \times 10^{0}$ | $1.72 \times 10^{0}$ |
| | | adacore | $1.31 \times 10^{0}$ | $1.29 \times 10^{0}$ | $1.39 \times 10^{0}$ | $1.38 \times 10^{0}$ | $1.40 \times 10^{0}$ | $1.72 \times 10^{0}$ |
| | | el2n | $1.61 \times 10^{0}$ | $1.53 \times 10^{0}$ | $1.66 \times 10^{0}$ | $1.53 \times 10^{0}$ | $1.58 \times 10^{0}$ | $1.72 \times 10^{0}$ |
| | | graNd | $1.75 \times 10^{0}$ | $1.59 \times 10^{0}$ | $1.57 \times 10^{0}$ | $1.48 \times 10^{0}$ | $1.52 \times 10^{0}$ | $1.72 \times 10^{0}$ |
| | PICore | craig | $1.37 \times 10^{0}$ | $1.55 \times 10^{0}$ | $1.51 \times 10^{0}$ | $1.59 \times 10^{0}$ | $1.51 \times 10^{0}$ | $1.72 \times 10^{0}$ |
| | | gradmatch | $1.76 \times 10^{0}$ | $1.59 \times 10^{0}$ | $1.56 \times 10^{0}$ | $1.53 \times 10^{0}$ | $1.49 \times 10^{0}$ | $1.72 \times 10^{0}$ |
| | | adacore | $1.22 \times 10^{0}$ | $1.29 \times 10^{0}$ | $1.25 \times 10^{0}$ | $1.30 \times 10^{0}$ | $1.54 \times 10^{0}$ | $1.72 \times 10^{0}$ |
| | | el2n | $1.67 \times 10^{0}$ | $1.59 \times 10^{0}$ | $1.59 \times 10^{0}$ | $1.54 \times 10^{0}$ | $1.52 \times 10^{0}$ | $1.72 \times 10^{0}$ |
| | | graNd | $1.77 \times 10^{0}$ | $1.57 \times 10^{0}$ | $1.59 \times 10^{0}$ | $1.55 \times 10^{0}$ | $1.52 \times 10^{0}$ | $1.72 \times 10^{0}$ |

Table 15: Test NRMSE on the Darcy dataset at resolution 128 across varying coreset percentages (20%–100%) between supervised and PICore-based coreset selection methods using both FNO and UNO architectures.

| Operator | Method | Algorithm | 20.0% | 30.0% | 40.0% | 60.0% | 80.0% | 100.0% |
|---|---|---|---|---|---|---|---|---|
| FNO | Supervised | craig | $2.13 \times 10^{-1}$ | $7.81 \times 10^{-2}$ | $7.24 \times 10^{-2}$ | $7.33 \times 10^{-2}$ | $7.05 \times 10^{-2}$ | $8.54 \times 10^{-2}$ |
| | | gradmatch | $1.57 \times 10^{-1}$ | $7.36 \times 10^{-2}$ | $7.24 \times 10^{-2}$ | $7.55 \times 10^{-2}$ | $7.49 \times 10^{-2}$ | $8.54 \times 10^{-2}$ |
| | | adacore | $3.18 \times 10^{-1}$ | $6.85 \times 10^{-2}$ | $6.16 \times 10^{-2}$ | $6.55 \times 10^{-2}$ | $6.48 \times 10^{-2}$ | $8.54 \times 10^{-2}$ |
| | | el2n | $1.43 \times 10^{-1}$ | $7.76 \times 10^{-2}$ | $7.38 \times 10^{-2}$ | $7.93 \times 10^{-2}$ | $7.64 \times 10^{-2}$ | $8.54 \times 10^{-2}$ |
| | | graNd | $1.67 \times 10^{-1}$ | $7.42 \times 10^{-2}$ | $6.78 \times 10^{-2}$ | $7.31 \times 10^{-2}$ | $6.98 \times 10^{-2}$ | $8.54 \times 10^{-2}$ |
| | PICore | craig | $1.73 \times 10^{-1}$ | $7.56 \times 10^{-2}$ | $6.50 \times 10^{-2}$ | $6.72 \times 10^{-2}$ | $6.75 \times 10^{-2}$ | $8.54 \times 10^{-2}$ |
| | | gradmatch | $1.70 \times 10^{-1}$ | $7.83 \times 10^{-2}$ | $7.36 \times 10^{-2}$ | $7.61 \times 10^{-2}$ | $7.76 \times 10^{-2}$ | $8.54 \times 10^{-2}$ |
| | | adacore | $2.84 \times 10^{-1}$ | $1.04 \times 10^{-1}$ | $6.38 \times 10^{-2}$ | $6.38 \times 10^{-2}$ | $6.90 \times 10^{-2}$ | $8.54 \times 10^{-2}$ |
| | | el2n | $1.63 \times 10^{-1}$ | $7.51 \times 10^{-2}$ | $7.04 \times 10^{-2}$ | $7.61 \times 10^{-2}$ | $7.65 \times 10^{-2}$ | $8.54 \times 10^{-2}$ |
| | | graNd | $1.55 \times 10^{-1}$ | $7.83 \times 10^{-2}$ | $7.57 \times 10^{-2}$ | $7.53 \times 10^{-2}$ | $7.75 \times 10^{-2}$ | $8.54 \times 10^{-2}$ |
| UNO | Supervised | craig | $5.99 \times 10^{-2}$ | $5.79 \times 10^{-2}$ | $5.64 \times 10^{-2}$ | $5.56 \times 10^{-2}$ | $5.52 \times 10^{-2}$ | $5.16 \times 10^{-2}$ |
| | | gradmatch | $5.99 \times 10^{-2}$ | $5.83 \times 10^{-2}$ | $5.68 \times 10^{-2}$ | $5.60 \times 10^{-2}$ | $5.53 \times 10^{-2}$ | $5.16 \times 10^{-2}$ |
| | | adacore | $6.45 \times 10^{-2}$ | $5.98 \times 10^{-2}$ | $5.77 \times 10^{-2}$ | $5.58 \times 10^{-2}$ | $5.49 \times 10^{-2}$ | $5.16 \times 10^{-2}$ |
| | | el2n | $5.93 \times 10^{-2}$ | $5.77 \times 10^{-2}$ | $5.66 \times 10^{-2}$ | $5.57 \times 10^{-2}$ | $5.52 \times 10^{-2}$ | $5.16 \times 10^{-2}$ |
| | | graNd | $5.94 \times 10^{-2}$ | $5.82 \times 10^{-2}$ | $5.69 \times 10^{-2}$ | $5.60 \times 10^{-2}$ | $5.52 \times 10^{-2}$ | $5.16 \times 10^{-2}$ |
| | PICore | craig | $6.07 \times 10^{-2}$ | $5.74 \times 10^{-2}$ | $5.66 \times 10^{-2}$ | $5.53 \times 10^{-2}$ | $5.51 \times 10^{-2}$ | $5.16 \times 10^{-2}$ |
| | | gradmatch | $5.97 \times 10^{-2}$ | $5.76 \times 10^{-2}$ | $5.67 \times 10^{-2}$ | $5.60 \times 10^{-2}$ | $5.55 \times 10^{-2}$ | $5.16 \times 10^{-2}$ |
| | | adacore | $6.56 \times 10^{-2}$ | $6.02 \times 10^{-2}$ | $5.81 \times 10^{-2}$ | $5.57 \times 10^{-2}$ | $5.50 \times 10^{-2}$ | $5.16 \times 10^{-2}$ |
| | | el2n | $5.96 \times 10^{-2}$ | $5.76 \times 10^{-2}$ | $5.68 \times 10^{-2}$ | $5.58 \times 10^{-2}$ | $5.53 \times 10^{-2}$ | $5.16 \times 10^{-2}$ |
| | | graNd | $5.96 \times 10^{-2}$ | $5.77 \times 10^{-2}$ | $5.65 \times 10^{-2}$ | $5.58 \times 10^{-2}$ | $5.54 \times 10^{-2}$ | $5.16 \times 10^{-2}$ |

Table 16: Test NRMSE on the Navier Stokes Incompressible dataset at resolution 256 across varying coreset percentages (20%–100%) between supervised and PICore-based coreset selection methods using both FNO and UNO architectures.

