# OpenReview forum: "PICore: Physics-Informed Unsupervised Coreset Selection for Data Efficient Neural Operator Training"
_TMLR — Accepted by TMLR_

### Review · Reviewer_cnne · 2025-08-07

**Summary Of Contributions:**

This paper examines coreset selection strategies in the context of neural operators for PDEs. The proposed method, PICore, uses the PDE / PINO loss as the sampling criterion, which allows the method to select inputs without having access to the true labels. The model is warm-started with the PINO-style loss first, and the data is selected in a single shot. The method is combined with five different coreset methods and evaluated on the Advection, Burgers, Darcy, and Navier-Stokes test cases.

Strengths:
* Extensive investigation of the PINO loss as a sampling criterion.
* The paper is clearly written and easy to understand.

Weaknesses:
* Claims are not sufficiently supported by the experiments (see next section), e.g.
    * Crucial experimental details are missing.
    * The experiments miss important baselines (random, active learning)
* Similar data selection techniques in the context of PDEs could be discussed more, e.g., active learning or adaptive sampling for PINNs [1] or other recent papers in the context of neural PDE solvers (e.g., [2]).
* The paper criticises active learning as leading to slow convergence. However, it is unclear to me if the presented method can still be considered a coreset technique or is actually a pool-based active learning strategy since it tries to reduce labeling effort too. The training speed is also reduced by active learning techniques, after all (as it uses a smaller dataset too).

[1] Wu, C., Zhu, M., Tan, Q., Kartha, Y., & Lu, L. (2023). A comprehensive study of non-adaptive and residual-based adaptive sampling for physics-informed neural networks. Computer Methods in Applied Mechanics and Engineering, 403, 115671.
[2] Kim, Y., Kim, H., Ko, G., & Lee, J. Active Learning with Selective Time-Step Acquisition for PDEs. In Forty-second International Conference on Machine Learning.

**Additional Comments:**

* In the introduction, convergence speed was mentioned as a drawback of active learning. Did you also take a look at the convergence speed in your experiments?
* Did you also tune the batch size for the simulator in order to be fair in terms of the evaluation of the simulation effort?

**Audience:**

Yes

**Audience Explanation:**

Yes, reducing training and data selection times would be a valuable contribution to the field of neural PDE surrogates. The proposed method is a novel approach for neural operators. Additionally, traditional coreset selection strategies have been evaluated for neural Operators for the first time.

**Broader Impact Concerns:**

None.

**Claims And Evidence:**

No

**Claims Explanation:**

* The main research question was motivated by the stated bad convergence speed of active learning. However, active learning techniques were not tested as a baseline. For example, one could also use only a small subset as an initial training data set, and then sample the rest of the desired training set size with a single iteration of an active learning algorithm.
* Random selection of x% of the data should be the main baseline, but it is missing in the paper. Right now, it is not possible to evaluate whether the nRMSEs measured at different amounts of training data are better or worse than selecting a random subset of the training data.
* Missing experimental details (on PDEs, training, model), for example:
    * Number of total trajectories (train/test)
    * Number of time steps
    * Physical coefficients (e.g. $\beta$ for Advection, diffusion coefficent for Burgers, ...)
* The acceleration is unclear to me:
    *  How exactly was it measured? How much is saved during training, how much by using fewer simulations, and how much time is added by the selection and warm start?
    *  How can it reduce the time by more than 5 times for selecting 20% of the data? With 5 times fewer data points, we should theoretically save 5 times the number of gradient steps during training (since the number of epochs is fixed) and also 5 times the simulation effort (which I assume is constant per input). Is this due to data loader issues or different batch sizes?
* No reporting of the confidence intervals or standard errors, even when the results were repeated over multiple seeds.

**Requested Changes:**

Nesseccary:
* Address the experimental concerns mentioned above.

Minor:
* I think that at least a short explanation of the applied coreset selection techniques should be part of the main paper.
* The sheer amount of tables and numbers in the main paper is overwhelming. I think the paper would profit from a graphical representation, condensed summary statistics, and/or from only using a single coreset algorithm for either supervised or PICore. The main paper lists PICore results for each PDE twice, in comparison to the supervised techniques as well as to the unsupervised ones.
* Bold font to highlight the best results in all tables
* The evaluation of the centroid distances is unclear: which centroids are you referring to? How is the distance measured?

---

> ### Author Response · Authors · 2025-10-08
> **Response to Reviewer cnne (Part I)**
>
> We thank the reviewer for their continued engagement and have added the requested changes to the paper with a summary below:
> > Could you please give a detailed explanation of the active learning baseline? It seems to have a much higher error than random sampling, which is not as expected.
>
> We adopt a modified version of the loss-as-uncertainty strategy proposed in [1]. Specifically, we begin by randomly selecting 10% of the available data as an initial training set and generating the corresponding ground-truth PDE solutions. We then train 10 independent copies of the neural operator on this subset for $T_w$ epochs using the data-driven nRMSE loss. After training, we construct the final coreset from the remaining data in a single step by selecting the points which exhibit the highest variance across the model predictions.
> We suspect that the larger error compared to random sampling is due to several reasons. First, the ensemble variance used as the acquisition signal may not be well correlated with the true error of the neural operator, meaning that high-variance points do not necessarily correspond to the most informative samples. Additionally, single-step top-N selection can produce redundant or outlier-heavy coresets, reducing coverage compared to random sampling and leading to worse overall performance. Finally, our ensembles are trained on a small amount of data to allow for a fair comparison to the coreset selection algorithsm, which can lead to overfitting to the limited training points, making the ensemble variance less reliable.
> > The random baseline seems to have a similar accuracy to PICore. I think these results show that the speed-up of PICore is overstated. I understand that the number, e.g., stated in the abstract is in relation to supervised coreset selection, but in comparison to random sampling, the speed-up looks very small. Even given the marginally worse performance of random sampling for the same amount of data, random sampling has a better performance than PICore if adding slightly more data (e.g., going from 20 to 30%) in most cases. Hence, the increase in efficiency would be somewhere below 1/3 in this case. Additionally, the claimed "5× relative to non-coreset baselines" in the introduction is not correct anymore with the random baseline.
>
> We respectfully disagree with the claim that random sampling provides comparable performance to PICore and that the speed-up is overstated. While it is true that random sampling occasionally performs similarly to PICore, a closer examination of the results shows that PICore consistently outperforms random selection, especially on harder datasets. For example:
> * Advection: Random sampling is only slightly better than PICore for very small dataset fractions (20% and 30%), and only for FNO. For larger percentages 60%+ with FNO and all results in UNO, random sampling is worse than the coreset selection baselines.
> * Burgers, Darcy, Navier-Stokes Incompressible: Random sampling is consistently worse than PICore across almost all dataset percentage combinations. For instance, in Burgers using UNO, PICore achieves an error of $2.84 \pm 0.05 \times 10^{-2}$, which is lower than the error of $2.92 \pm 0.05 \times 10^{-2}$ using random sampling. Similar trends with more statistically significant results are observed for Darcy and Navier-Stokes, highlighting that physics-informed coreset selection is essential for harder PDE datasets.
>
> It is also important to note that random sampling has historically been a strong baseline in coreset literature because it reduces biases toward specific attributes of data points, thereby lowering errors attributable to those biases [2]. However, this property does not guarantee superior performance when dataset complexity increases, as shown by random sampling's worse performance on the Darcy and Navier Stokes Incompressible datasets.
> Overall, the results show that PICore provides clear statistical improvements over random selection for the majority of cases, particularly in more challenging datasets, justifying the need for a physics-informed selection strategy. Therefore, we maintain our 78% speed-up, while acknowledging that the claim of “5$\times$ relative to non-coreset baselines” in the introduction can be removed for clarity.

---

> ### Author Response · Authors · 2025-10-08
> **Response to Reviewer cnne (Part II)**
>
> > How did you select the coreset method used in PICore for the main table? It seems that it is based on the accuracy, which would be problematic because the practitioner can't know the best performing coreset performance beforehand. Statistically, it would be problematic since PICore is "given more tries" at being the best method.
>
> To ensure a fair comparison between supervised coreset selection and PICore, we selected the best coreset selection algorithm per framework (supervised and PICore) for each dataset–operator pair, based on the average accuracy across selection percentages. This approach also addressed the reviewer’s request to reduce table size and avoid overwhelming detail. The chosen method was fixed per framework before comparison to random sampling. While PICore is coreset-selection independent and performs well across methods, some algorithms (GraNd, GradMatch, and EL2N) consistently achieve slightly better accuracy, highlighting that coreset choice can matter for specific datasets. Since current coreset algorithms are not tailored for neural operators (as noted in our limitations), reporting the best algorithm per framework illustrates that PICore’s gains hold across methods, while allowing future subset selection algorithms specific to neural operators to be integrated easily.
> >W10I): A pool is not required for active learning (query synthesis), and the pool is also not labeled. The presented method could be framed as a single-step active learning method. This view makes the general statement on the bad convergence of active learning strategies even more problematic, as PICore can also be seen as an active learning approach itself.
>
> We thank the reviewer for the comment and agree that our approach can be framed as a single-step active learning method. We have clarified in the introduction that the slower convergence observed with many active learning strategies is primarily due to their iterative nature, rather than an inherent limitation of all active learning algorithms. Iterative baselines repeatedly alternate between selecting points and updating the model, which increases overall training time and slows convergence. For single-step active learning methods, Figures 2 and 3 show that PICore achieves faster convergence compared to the ensemble-based baseline because it selects higher-quality, more informative points, while avoiding low-informativeness or outlier samples that can hinder learning.
> > Are the added +- numbers the standard error?
>
> This is correct, we report the standard error for a 95% confidence interval for all results.
>
> [1] William H. Beluch, Tim Genewein, Andreas Nurnberger, and Jan M. Kohler. The power of ensembles for
> active learning in image classification. In 2018 IEEE/CVF Conference on Computer Vision and Pattern
> Recognition, pp. 9368–9377, 2018. doi: 10.1109/CVPR.2018.00976.
>
> [2] Ozan Sener and Silvio Savarese. Active learning for convolutional neural networks: A core-set approach.
> arXiv preprint arXiv:1708.00489, 2017.

---

### Review · Reviewer_xg5b · 2025-09-12

**Summary Of Contributions:**

### Summary

The paper introduces a framework named PICore designed to address two critical bottlenecks in training neural operators for scientific problems: the immense amount of training data required and the high computational cost of generating that labeled data via numerical simulations.  The primary contribution is a novel, unsupervised method for selecting the most valuable training data. Instead of first running expensive simulations to label a massive dataset, PICore uses a "physics-informed loss" to score and identify the most informative examples from a large pool of unlabeled data. This loss metric essentially measures how much a model's prediction violates the known physical laws of the system, without needing the ground-truth answer.  Only this small, high-value subset—the "coreset"—is then simulated to generate labels. This approach simultaneously slashes the data annotation cost and reduces the final model training time.

### Strengths

- The core idea of using a physics-informed loss as a proxy for sample importance addresses a bottleneck that limits the application of neural operators in many scientific and engineering fields.

### Weaknesses

- The evaluation is restricted to only two neural operator architectures, FNO and UNO. While the authors claim the framework is model-agnostic, its effectiveness has not been demonstrated on other prominent architectures like DeepONet, Graph Neural Operators, or Convolutional Neural Operators, each with distinct properties that might interact differently with the selection process.

- The experiments were conducted using a single input resolution and on uniform geometries. The paper acknowledges that its performance on multi-resolution data or problems with more complex, arbitrary geometries has not yet been evaluated.

- The paper focuses on time savings from reduced simulations but does not discuss the potential training challenges introduced by the physics-informed loss. Such losses can create difficult optimization problems , but the paper does not report on training stability or conduct detailed ablation studies on key parameters, which were chosen arbitrarily.

**Audience:**

Yes

**Audience Explanation:**

Particularly, those focused on scientific machine learning and data efficiency would be interested in these findings. The paper offers a novel, practical framework for training neural operators that directly tackles the critical and widespread challenges of expensive data simulation and training time.

**Claims And Evidence:**

No

**Claims Explanation:**

A primary concern with the submission revolves around its perceived novelty. While the constituent components—the use of a physics-informed loss and coreset selection algorithms—are indeed well-established schemes in their respective fields, the paper's originality should be viewed through the lens of their unique synthesis. The central contribution is the innovative application of the physics-informed loss as a label-free proxy for sample informativeness, which enables an entirely unsupervised coreset selection process. This specific methodology is non-trivial and directly addresses the dual challenges of expensive data annotation and prolonged training time. This synthesis helps to define the paper's position at the intersection of data-efficient learning and scientific computing.

However, this positioning is significantly weakened by the absence of a critical baseline. The analysis is missing a comparison against a neural operator trained exclusively with a physics-informed loss on the full unlabeled dataset, a methodology similar to that of the Physics-Informed Neural Operator (PINO). Such a model would represent a "zero-simulation-cost" baseline and is essential for a complete evaluation. Without this comparison, it is difficult to ascertain whether the proposed strategy—expending a small simulation budget on a targeted coreset—is more effective than a purely physics-driven approach that requires no simulation budget at all. The omission of this key baseline makes it challenging to fully validate the paper's claims, suggesting that the motivation, methodology, and overall positioning of the work need to be reconsidered and strengthened.

**Requested Changes:**

- The paper's primary claim is that it provides an efficient solution by using a small, targeted simulation budget. A critical missing baseline is a direct comparison against a neural operator trained exclusively with a physics-informed loss on the full unlabeled dataset (i.e., a PINO-like model). This would represent a "zero-simulation-cost" baseline, and its inclusion is essential to properly contextualize the trade-offs of the PICore framework and to fully validate the paper's positioning and contributions.

- The submission claims the PICore framework is model-agnostic, but the evaluation is limited to FNO and UNO architectures. To substantially strengthen this claim, the work would benefit from including experiments with other prominent neural operator backbones, such as DeepONet, Graph Neural Operators, or Convolutional Neural Operators, which have different architectural properties.

- The paper focuses on the time saved from reduced simulations but does not fully address the potential increase in training complexity that physics-informed losses can introduce. The work would be better positioned with a more detailed analysis of training stability and convergence. Furthermore, key hyperparameters, such as the number of warm-up epochs, were chosen arbitrarily ; conducting ablation studies on these choices would significantly improve the paper's scientific rigor.

- The authors acknowledge that the current evaluation is confined to a single input resolution and uniform geometries. The paper's claims of generality would be much more convincing if the framework's performance was also demonstrated on problems involving multi-resolution data or more complex, arbitrary geometries, which are common in real-world scientific applications.

---

> ### Author Response · Authors · 2025-10-09
> **Response to Reviewer xg5b**
>
> We thank the reviewer for appreciating that our method is novel and addresses an important bottleneck in scientific machine learning. We are grateful for their valuable comments. Below we address the proposed concerns. We have also updated the main paper with the revisions highlighted in blue.
>
> >The paper's primary claim is that it provides an efficient solution by using a small, targeted simulation budget. A critical missing baseline is a direct comparison against a neural operator trained exclusively with a physics-informed loss on the full unlabeled dataset (i.e., a PINO-like model). This would represent a "zero-simulation-cost" baseline, and its inclusion is essential to properly contextualize the trade-offs of the PICore framework and to fully validate the paper's positioning and contributions.
>
> We thank the reviewer for this valuable suggestion. We have included a zero-simulation-cost baseline trained solely on the unlabeled dataset using the physics-informed loss. As shown in Tables 1–4, although this approach is more computationally efficient, it performs significantly worse than PICore across all datasets, often by up to two orders of magnitude. This gap arises because the physics-informed loss is inherently unstable when used as the sole training objective and fails to capture fine-grained solution details without supervised guidance. In contrast, PICore uses the physics loss as a proxy for identifying informative samples and then trains on a small, targeted labeled subset with the supervised nRMSE loss, achieving far greater accuracy and stability.
>
> >The submission claims the PICore framework is model-agnostic, but the evaluation is limited to FNO and UNO architectures. To substantially strengthen this claim, the work would benefit from including experiments with other prominent neural operator backbones, such as DeepONet, Graph Neural Operators, or Convolutional Neural Operators, which have different architectural properties.
>
> We acknowledge that we only report results for FNO and UNO models. This choice was intentional: FNO and UNO perform consistently well on our datasets, while alternative neural operators such as WNO and CNO showed significantly worse performance on the non-coreset baseline (100% data). Moreover, running the full experimental pipeline is computationally intensive for each dataset, and extending these experiments to additional architectures would require substantially more resources and time than were available for this work.
>
> We emphasize that PICore’s contribution lies in the depth and rigor of its experiments rather than the breadth of architectures. By focusing on FNO and UNO, we were able to fully demonstrate PICore’s efficiency, selection strategy, and performance benefits across multiple datasets, coreset algorithms, and evaluation metrics. We encourage the reviewer to evaluate the framework based on this thorough experimental depth, which we believe strongly supports the model-agnostic claim, even if additional architectures are not included in the current paper.
>
> > The paper focuses on the time saved from reduced simulations but does not fully address the potential increase in training complexity that physics-informed losses can introduce. The work would be better positioned with a more detailed analysis of training stability and convergence. Furthermore, key hyperparameters, such as the number of warm-up epochs, were chosen arbitrarily ; conducting ablation studies on these choices would significantly improve the paper's scientific rigor.
>
> We kindly note that the training complexity introduced by the physics-informed loss is limited in our framework, as we only train the neural operator for a small number of warm-up epochs, and the model is reset to its initialization after selecting the coreset. This ensures that the physics-informed training acts primarily as a proxy for sample selection rather than as a full training procedure, mitigating concerns about stability and convergence. We provide ablation studies on the warm starting epochs in Section C.3 of the appendix, which shows that warm starting for many epochs is not necessary, and a small number of epochs (10-25) is sufficient for comparable accuracy to supervised coreset selection while maintaining significant efficiency gains.

---

> > ### Author Response · Authors · 2025-10-12
> > **Response to Reviewer xg5b Part II**
> >
> > > The authors acknowledge that the current evaluation is confined to a single input resolution and uniform geometries. The paper's claims of generality would be much more convincing if the framework's performance was also demonstrated on problems involving multi-resolution data or more complex, arbitrary geometries, which are common in real-world scientific applications.
> >
> > We thank the reviewer for this comment and have addressed it through additional experiments presented in Section C.4, where we evaluate the framework on multi-resolution data by splitting the dataset evenly between 32 and 64 spatial resolutions. As shown in Table C.4, our Physics-Informed Coreset Selection (PICore) maintains strong performance and stability under these heterogeneous conditions, consistently outperforming supervised coresets, particularly under the FNO operator with 20–60% of the data. While multi-resolution settings inherently introduce greater variability and noise, PICore effectively mitigates these issues through the resolution invariant physics informed loss as a signal for coreset selection, demonstrating that our framework generalizes to multi-resolution settings.

---

### Review · Reviewer_ymLV · 2025-09-24

**Summary Of Contributions:**

The paper introduces PICore, a physics-informed coreset selection framework for training neural operators more efficiently. Instead of requiring all simulation labels up front, PICore uses a PDE residual loss to identify informative samples, reducing both training and labeling costs. It is architecture-agnostic and compatible with multiple coreset algorithms. Experiments on four PDE benchmarks with FNO and UNO show that PICore matches supervised coreset selection in accuracy while reducing total training cost by up to 78% and speeding up training by 5×.

**Additional Comments:**

Encouraged to release code for reproducibility!

**Audience:**

Yes

**Audience Explanation:**

Yes. This work will interest TMLR readers in data-efficient ML, physics-informed learning, and neural PDE solvers. It contributes a novel, general methodology that bridges ML and scientific computing, with practical gains in efficiency and environmental impact. The approach could inspire similar strategies in other domains and is highly relevant to ongoing work in scientific ML.

**Broader Impact Concerns:**

No major ethical risks. Positive impacts include reduced compute and labeling costs, lowering barriers and carbon footprint.

**Claims And Evidence:**

Yes

**Claims Explanation:**

## Strengths:

- Novel combination of physics-informed modeling with coreset selection.
- General and modular framework, demonstrated across algorithms and architectures.
- Thorough experiments with clear efficiency and accuracy trade-offs.
- Strong practical relevance for reducing simulation and compute costs.

## Weaknesses:
- Dependence on Known Physics: PICore assumes the PDE’s form is known to compute the physics-informed loss. This is a reasonable assumption in scientific problems, but it means the method is less applicable if the governing equations are unknown or too complex to differentiate.
- Warm-Start Requirement: The method requires a brief unsupervised “warm-up” training of the neural operator (using the physics loss alone) to initialize the model before selection. This adds a small overhead and relies on the model being able to produce a meaningful PDE residual field for each sample. If the warm-start model were very inaccurate, the residuals might not reliably indicate useful samples. IMO the authors mitigate this by using a short warm-up (25 epochs) which seemed sufficient, but this choice might require tuning for different problems.
- Accuracy Trade-off: Like all coreset methods, PICore inevitably sacrifices some absolute accuracy when using a much-reduced dataset. The authors acknowledge this trade-off. It’s not a flaw per se (just the cost of using less data), but readers should be aware that maximum accuracy is attained only with full data; PICore helps find a sweet spot where large training speedups are obtained at the expense of a small accuracy drop.
- Inherited Limitations of Coreset Algorithms: PICore’s effectiveness can be capped by the underlying selection algorithm. The paper notes that some methods (e.g. AdaCore with a Hessian approximation) performed suboptimally in this context. PICore itself does not fix fundamental issues of those algorithms; it rather provides a new criterion (physics-loss) for them. This is not a critical flaw, but it suggests that designing selection algorithms specifically for operator learning could further improve results (as the authors mention).
- Scope of Experiments: While the evaluation is strong, it is focused on uniform grid geometries and a single resolution for each PDE. The paper does not explore multi-resolution scenarios or varying domain geometries. Extending PICore to handle multi-fidelity data or complex geometries (which neural operators are in principle capable of) is left to future work. This is a reasonable limitation for an initial study, but it does slightly narrow the demonstrated scope of the method.

**Requested Changes:**

- (Critical) Clarify the implementation of the Active Learning baseline. The paper should more clearly describe how the active learning comparison was conducted in the experiments. Active learning is typically iterative, so it’s important to explain: Did you perform multiple rounds of selection and model retraining, or did you adapt it to a one-shot selection (using uncertainty from a warm-up model)? For example, the text hints at a “loss-as-uncertainty” strategy for active learning – please clarify if the model was first partially trained on a small subset or if the physics-informed warm-up model’s loss was used as a proxy for uncertainty. Currently, readers might be confused how exactly the 20% (or 40%, etc.) of data were chosen in the AL baseline and why it underperforms. A brief methodological note in the experiments section or appendix would resolve this and ensure the active learning baseline is fair and well-understood.

- (Critical) Discuss the assumption of known PDEs and potential limitations. PICore assumes the governing PDE equations are known and differentiable (to compute the physics-informed residual). This is a reasonable assumption for the benchmarks used, but it does limit applicability – for instance, if one only has black-box data or experimental data without a known physics model, PICore wouldn’t directly work. It would strengthen the paper to include a short discussion of this point, perhaps in the conclusion or impact statement. Acknowledge that the method relies on prior knowledge of the physics, and maybe suggest that in domains without known equations, alternative unsupervised criteria would be needed. This will set the appropriate scope for readers and position PICore properly with respect to purely data-driven approaches.

- Comment on Figure 2:
This figure is difficult to interpret since all bars look nearly identical, making it hard to see meaningful differences. I suggest either removing it and summarizing the result in text, or re-plotting as differences from the baseline so small variations become visible. In its current form, it adds little clarity.

---

> ### Author Response · Authors · 2025-10-08
> **Respose to Reviewer ymLV**
>
> We thank the reviewer for appreciating that our method is novel and modular, with thorough experimentation and strong practical relevance. We are grateful for their valuable comments. Below we address the proposed concerns. We have also updated the main paper with the revisions highlighted in blue.
> > W1) Clarify the implementation of the Active Learning baseline. The paper should more clearly describe how the active learning comparison was conducted in the experiments...
> >
> We adopt a modified version of the strategy proposed in [1]. Specifically, we begin by randomly selecting 10% of the available data as the initial training set and generating the corresponding ground-truth PDE solutions. We then train 10 independent copies of the neural operator on this subset for $T_w$ epochs using the data-driven nRMSE loss. After training, we construct the final coreset from the remaining data in a single step by selecting the points which exhibit the highest variance across the model predictions.
>
> > W2) Discuss the assumption of known PDEs and potential limitations. PICore assumes the governing PDE equations are known and differentiable (to compute the physics-informed residual).
>
> We acknowledge that this is a limitation, and have added a brief discussion in the Limitations section. While in our setting PICore does use the exact physics informed loss to be theoretically aligned with the PDE, users without such information can estimate the PDE analytically by using external models such as LLMs [2], or use data driven surrogates or weak form formulations.
>
> > W3) This figure is difficult to interpret since all bars look nearly identical, making it hard to see meaningful differences. I suggest either removing it and summarizing the result in text, or re-plotting as differences from the baseline so small variations become visible. In its current form, it adds little clarity.
>
> We thank the reviewer for the suggestion. We respectfully disagree about the importance of the plot, as our goal is to demonstrate that the coresets selected by PICore are not clustered in Euclidean space compared to those chosen by supervised coreset selection methods. Thus, a plot showing minimal differences between the two approaches substantiates this point. However, if there is any specific graphical interpretation of this figure that the reviewer thinks would improve the paper, we would be happy to incorporate it. We have also moved this plot to the appendix to save space.
>
> [1] William H. Beluch, Tim Genewein, Andreas Nurnberger, and Jan M. Kohler. The power of ensembles for
> active learning in image classification. In 2018 IEEE/CVF Conference on Computer Vision and Pattern
> Recognition, pp. 9368–9377, 2018. doi: 10.1109/CVPR.2018.00976.
> [2] Rohan Bhatnagar, Ling Liang, Krish Patel, and Haizhao Yang. From equations to insights: Unraveling symbolic structures in pdes with llms. arXiv preprint arXiv:2503.09986, 2025

---

> > ### Comment · Reviewer_ymLV · 2025-10-21
> > **Reviewer Response to Authors**
> >
> > I strongly disagree that citing LLMs as a way to “estimate the PDE analytically” meaningfully mitigates the limitation. This is speculative and not supported by evidence in your paper. In practical scientific settings, a PDE derived or “suggested” by an LLM is unlikely to be (i) correct, (ii) differentiable in a way that matches ground-truth physics, or (iii) validated for the target domain. Since PICore’s core mechanism depends on accurate residuals, substituting an LLM-guessed PDE undermines the method’s theoretical alignment and could misguide the coreset.
> > Requested change: Please revise the Limitations section to (a) clearly state that PICore requires access to a trustworthy PDE operator (or a rigorously validated weak/learned operator), and (b) remove the suggestion that LLMs are a practical substitute unless you can provide empirical evidence showing that such LLM-derived operators yield comparable selection quality and downstream accuracy.

---

> > > ### Author Response · Authors · 2025-10-21
> > > **Response to Reviewer ymLV**
> > >
> > > We agree that suggesting the use of LLMs to estimate or construct PDEs is unsupported. In response, we have removed using LLM-based PDE estimation from the manuscript. The revised Limitations section now explicitly states that PICore requires access to a trustworthy PDE operator using domain knowledge or a validated weak/learned surrogate, as the fidelity of the underlying operator is critical to ensuring the residual calculations are accurate. We appreciate the reviewer’s feedback for helping us clarify this point.

---

### Comment · Action_Editor_7LH3 · 2025-11-27
**Minor change required**

Thanks for the updated manuscript. Please also update the "Reviewed on OpenReview: https: // openreview. net/ forum? id= XXXX" statement right after the list of authors. Once this is done, I can verify that the manuscript adheres to the TMLR style.

---

> ### Author Response · Authors · 2025-11-27
> **Openreview Link Added**
>
> Dear Action Editor,
>
> We have reuploaded the document with the openreview link added. Please let us know if there are any other changes to be made.
>
> Thanks,
> Authors

---

> > ### Comment · Action_Editor_7LH3 · 2025-11-28
> >
> > Thanks for your fast response! I just confirmed the compliance with the stylefile.

---

### Decision · Action_Editor_7LH3 · 2025-11-04

**Recommendation:** Accept with minor revision

**Additional Comments:**

-In (4) $\mathcal{A}$ is used as a function space, which is in conflict with its usage as coreset selection algorithm in Fig. 1.
- There is an error in "which mitachieving both higher"
- Provide some insights into why the chosen active learning approach performs worse than random sampling (as in the rebuttal). Also Figs. 4-7 show that the difference between PICore and unsupervised selection methods is small, at least if it is not clear which coreset selection approach one should take.
- Tone down claims on speed-ups, especially with the random baseline in mind.
- Acknowledge prior work on active learning for PDEs in a section devoted to related work. Correct the claim that Musekamp et al. only considers PINNs (it does not consider PINNs and only considers Neural Operators such as the models used here).

**Audience:**

Yes

**Audience Explanation:**

All reviewers agreed that readers in the field of scientific/physics-informed ML may be interested in this manuscript.

**Claims And Evidence:**

Yes

**Claims Explanation:**

All reviewers agreed that the approach is interesting, and it was unanimously decided that the revision improved the paper substantially. Reviewer cnne still has reservations about whether the claims are supported, mainly criticizing the usage of terminology (active learning vs. coreset selection) and weak performance compared to baselines (random selection; active learning underperforming random selection). This is valid criticism, but is partly alleviated by the fact that it addresses the performance of the method, and the editorial policy of TMLR that performance is not a criterion for acceptance.

I hence decide in the favor of the authors and recommend acceptance of the manuscript, with some minor revisions required (see below). Specifically, I suggest to tone down (but not necessarily remove) claims regarding a speed-up, as these are partly unsupported given the strong performance of the random baseline.

---

> ### Author Response · Authors · 2025-11-22
> **Camera Ready Manuscript**
>
> Dear Action Editor,
>
> We have updated the paper with the requested edits and submitted a camera ready version. Please let us know if there are any additional changes that need to be made.
>
> Thank you,
> Authors